# Estimating dispersal rates and locating genetic ancestors with genome-wide genealogies

Matthew Osmond[1]*, Graham Coop[2]*

[1]Department of Ecology and Evolutionary Biology, University of Toronto, Toronto, Canada; [2]Department of Evolution & Ecology and Center for Population Biology, University of California, Davis, Davis, United States

**Abstract** Spatial patterns in genetic diversity are shaped by individuals dispersing from their parents and larger-scale population movements. It has long been appreciated that these patterns of movement shape the underlying genealogies along the genome leading to geographic patterns of isolation-by-distance in contemporary population genetic data. However, extracting the enormous amount of information contained in genealogies along recombining sequences has, until recently, not been computationally feasible. Here, we capitalize on important recent advances in genome-wide gene-genealogy reconstruction and develop methods to use thousands of trees to estimate per-generation dispersal rates and to locate the genetic ancestors of a sample back through time. We take a likelihood approach in continuous space using a simple approximate model (branching Brownian motion) as our prior distribution of spatial genealogies. After testing our method with simulations we apply it to *Arabidopsis thaliana*. We estimate a dispersal rate of roughly 60 km$^2$/generation, slightly higher across latitude than across longitude, potentially reflecting a northward post-glacial expansion. Locating ancestors allows us to visualize major geographic movements, alternative geographic histories, and admixture. Our method highlights the huge amount of information about past dispersal events and population movements contained in genome-wide genealogies.

*For correspondence:
mm.osmond@utoronto.ca (MO);
gmcoop@ucdavis.edu (GC)

## Editor's evaluation

This fundamental and pioneering paper demonstrates the power of using the Ancestral Recombination Graph in estimating historical dispersal rates and illustrates the importance of using good data. The methodology is compelling and well beyond the state-of-the-art. The paper should be of interest to anyone working with population genetic inference.

## Introduction

Patterns of genetic diversity are shaped by the movements of individuals, as individuals move their alleles around the landscape as they disperse. Patterns of individual movement reflect individual-level dispersal; children move away from their parents' village and dandelion seeds blow in the wind. These patterns also reflect large-scale movements of populations. For example, in the past decade we have learnt about the large-scale movement of different peoples across the world from ancient DNA (*Slatkin and Racimo, 2016*; *Reich, 2018*). Such large-scale movements of individuals also occur in other species during biological invasions or with the retreat and expansion of populations in and out of glacial refugia, tracking the waxing and waning of the ice ages (*Hewitt, 2000*).

An individual's set of genealogical ancestors expands rapidly geographically back through time in sexually reproducing organisms (*Kelleher et al., 2016b*; *Coop, 2017*). Due to limited recombination

each generation, more than a few tens of generations back an individual's genetic ancestors represent only a tiny sample of their vast number of genealogical ancestors (*Donnelly, 1983*; *Coop, 2013*). Yet the geographic locations of genetic ancestors still represent an incredibly rich source of information on population history (*Bradburd and Ralph, 2019*). We can hope to learn about the geography of genetic ancestors because individuals who are geographically close are often genetically more similar across their genomes; their ancestral lineages have only dispersed for a relatively short time and distance since they last shared a geographically close common ancestor. This pattern is termed isolation-by-distance. These ideas about the effects of geography and genealogy have underlain our understanding of spatial population genetics since its inception (*Wright, 1943*; *Malécot, 1948*). Under coalescent models, lineages move spatially, as a Brownian motion if dispersal is random and local, splitting to give rise to descendent lineages until we reach the present day. Such models underlie inferences based on increasing allele frequency differentiation (such as $F_{ST}$) with geographic distance (*Rousset, 1997*) and the drop-off in the sharing of long blocks of genome shared identical by descent among pairs of individuals (*Ralph and Coop, 2013*; *Ringbauer et al., 2017*). These models also are the basis of methods that seek violations of isolation-by-distance (*Wang and Bradburd, 2014*).

While spatial genealogies have proven incredibly useful for theoretical tools and intuition, with few exceptions they have not proven useful for inferences because we have not been able to construct these genealogies along recombining sequences. In non-recombining chromosomes (e.g., mtDNA and Y), constructed genealogies have successfully been used to understand patterns of dispersal and spatial spread (*Avise, 2009*). However, these spatial genealogy inferences are necessarily limited as a single genealogy holds only limited information about the history of populations in a recombining species (*Barton and Wilson, 1995*). Phylogenetic approaches to geography 'phylogeography'; *Knowles, 2009* have been more widely and successfully applied to pathogens to track the spatial spread of epidemics, such as SARS-CoV-2 (*Martin et al., 2021*), but such approaches have yet to be applied to the thousands of genealogical trees that exist in sexual populations.

Here, we capitalize on the recent ability to infer a sequence of genealogies, with branch lengths, across recombining genomes (*Rasmussen et al., 2014*; *Speidel et al., 2019*; *Wohns et al., 2021*; *Schaefer et al., 2021*; *Zhang et al., 2023*; *Gunnarsson et al., 2024*; *Deng et al., 2024*, reviewed in *Wong et al., 2024*; *Lewanski et al., 2024*; *Nielsen et al., 2024*). Equipped with this information, we develop a method that uses a sequence of trees to estimate dispersal rates and locate genetic ancestors in continuous, two-dimensional space under the assumption of Brownian motion. Using thousands of approximately unlinked trees, we multiply likelihoods of the dispersal rate across trees to get a genome-wide estimate and use the sequence of trees to predict a cloud of ancestral locations as a way to visualize geographic ancestries. We first test our approach with simulations and then apply it to *Arabidopsis thaliana*, a species with a wide geographic distribution and a complex history (*Fulgione and Hancock, 2018*).

## Results
### Overview of approach
We first give an overview of our approach, the major components of which are illustrated in *Figure 1*. See Materials and methods for more details.

The rate of dispersal, which determines the average distance between parents and offspring, is a key parameter in ecology and evolution. To estimate this parameter we assume that in each generation the displacement of an offspring from its mother is normally distributed with a mean of 0 in each dimension and covariance matrix $\Sigma$. In two dimensions, the covariance matrix is determined by the standard deviations, $\sigma_x$ and $\sigma_y$, along the $x$ (longitudinal) and $y$ (latitudinal) axes, respectively, as well as the correlation, $\rho$, in displacements between these two axes. The average distance between a mother and its offspring is then $\sqrt{2/\pi}\sigma_i$ in each dimension.

Given this model, the path of a lineage from its ancestral location to the present day location is described by a Brownian motion with rate $\Sigma$. We therefore refer to $\Sigma$ as the dispersal rate, which describes the rate of increase in the variance of a location along a single lineage. Lineages covary in their locations because of shared evolutionary histories – lineages with a more recent common ancestor covary more. Given a tree at a locus we can calculate the covariance matrix of shared evolutionary times (back to the root) and compute the likelihood of the dispersal rate, $\Sigma$, which is normally

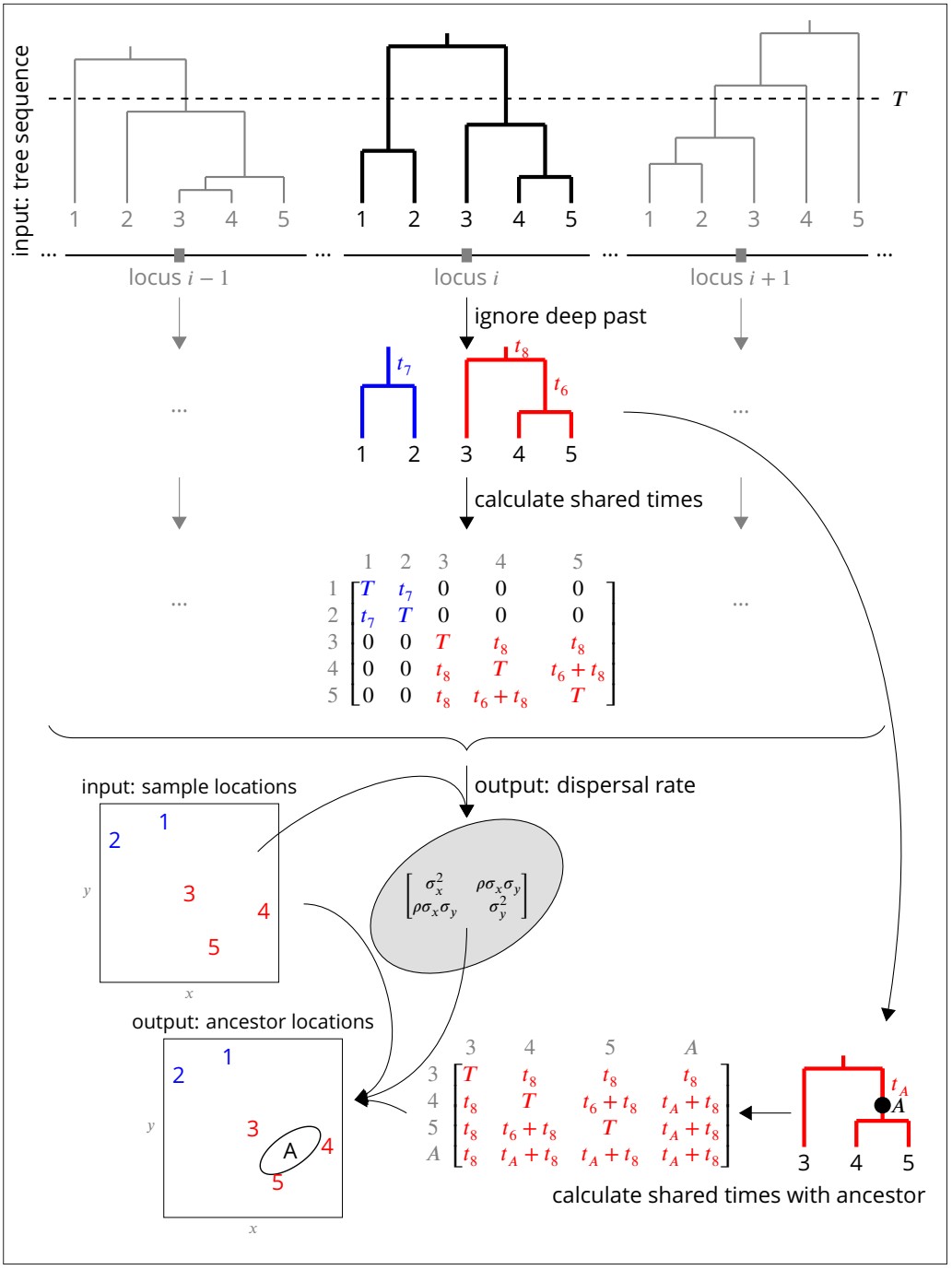

**Figure 1.** Conceptual overview of the approach. From a sequence of trees covering the full genome, we downsample to trees at approximately unlinked loci. To avoid the influence of strongly non-Brownian dynamics at deeper times (e.g., glacial refugia, boundaries), we ignore times deeper than $T$, which divides each tree into multiple subtrees (here, blue and red subtrees at locus $i$). From these subtrees, we extract the shared times of each pair of lineages back to the root. In practice (but not shown here), we use multiple samples of the tree at a given locus, for importance sampling, and also extract the coalescence times for importance sample weights. Under Brownian motion, the shared times describe the covariance we expect to see in the locations of our samples, and so using the times and locations we can find the maximum likelihood dispersal rate (a 2 × 2 covariance matrix). While we can estimate a dispersal rate at each locus, a strength of our approach is that we combine information across many loci, by multiplying likelihoods, to estimate a single genome-wide dispersal rate. Finally, we locate a genetic ancestor at a particular locus (a point on a tree, here $A$) by first calculating the time this ancestor shares with each of the samples in its subtree, and then using the shared times and dispersal rate to calculate the

*Figure 1 continued on next page*

*Figure 1 continued*

probability distribution of the ancestor's location conditioned on the sample locations. In practice (but not shown here), we calculate the location of the ancestor of a given sample at a given time across many loci, combining information across loci into a distribution of genome-wide ancestry across space.

distributed given this covariance matrix. At each locus we can estimate the likelihood of the dispersal rate given the tree at that locus allowing us to multiply likelihoods across loci to derive a genome-wide likelihood, and thus a genome-wide maximum likelihood dispersal rate. While the genome-wide dispersal estimate is not biased by correlations between the trees, we simply use every $n^{\text{th}}$ locus across the genome to reduce the redundancy of information we estimate dispersal from.

Under this same model we can also estimate the locations of genetic ancestors at a locus. Any point along a tree at any locus is a genetic ancestor of one or more current day samples. This ancestor's lineage has dispersed away from the location of the most recent common ancestor of the samples, and covaries with current day samples in their geographic location to the extent that it shares times in the tree with them. Under this model, the location of an ancestor is influenced by the locations of all samples in the same tree, including those that are not direct descendants (cf. *Wohns et al., 2021*). For example, in *Figure 1* the ancestor's location is not the midpoint of its two descendants (samples 4 and 5); the ancestor's location is also pulled towards sample 3 since the ancestor and sample 3 both arose from a common ancestor. Conditioning on the sample locations, and given the shared times and previously inferred dispersal rate, we can compute the probability the ancestor was at any location, which again is a normal distribution. In contrast to dispersal, for ancestral locations we do not want to multiply likelihoods across loci since the ancestors at distant loci are likely distinct. Instead, we calculate the maximum likelihood location of the genetic ancestor at each sparsely sampled locus to get a cloud of likely ancestral locations genome-wide, and use these clouds to visualize the spatial spread of genetic ancestry backwards through time.

We estimate marginal trees along the genome using Relate (*Speidel et al., 2019*). While we could in principle estimate trees independently at regions across the genome, a benefit of Relate and similar methods is that they estimate each tree using information from nearby regions. Relate infers a sequence of tree topologies and associated branch lengths, and can return a set of posterior draws of the branch lengths on a given tree (it was the only method that did so for a large number of samples when we began this work, but now see *Wohns et al., 2021*; *Deng et al., 2024*). This posterior distribution of branch lengths is useful to us as the shared times in the tree are key to the amount of time that individual lineages have had to disperse away from one another and we wish to include uncertainty in the times into our method. Relate gives us a posterior distribution of branch lengths that is estimated using a coalescent prior, which assumes a panmictic population of varying population size (the size changes are estimated as part of the method), where any two lineages are equally likely to coalesce. This panmictic prior results in a bias in the coalescent times under a spatial model, where geographically proximate samples are more likely to coalesce. To correct for this bias we make use of importance sampling to weight the samples of branch lengths at each locus. We then calculate the weighted average likelihood over our draws of our sample of trees at a locus (or loci), so that it is as if they were drawn from a prior of branching Brownian motion (*Meligkotsidou and Fearnhead, 2007*).

In practice, we concentrate on the recent past history of our sample. For our estimates of dispersal rates in particular we do not want to assume that our model of Gaussian dispersal (and branching Brownian motion) holds deep into the past history of the sample. This is because the long-term movement of lineages is constrained by geographic barriers (e.g., oceans) and larger-scale population movements may erase geographic signals over deep time scales. On a theoretical level ignoring the deep past may also be justified because in a finite habitat the locations of coalescence events further back in time become independent of sampling locations as lineages have moved around sufficiently (*Wilkins and Wakeley, 2002*). Thus we only use this geographic model to some time point in the past ($T$), and at each locus we use the covariance of shared branch lengths based on the set of subtrees formed by cutting off the full tree $T$ generations back. There are many ways one could choose the cutoff time, for example, the time to cross the habitat, the time since glaciation, or the most recent time at which the average number of lineages remaining across trees is less than a threshold. Here, we estimate dispersal across a range of $T$, which gives us a sense of how the effective dispersal rate changes over time. We then use the full trees ($T \to \infty$) to locate ancestors, so that we use all the

relatedness information. However, in our empirical application we only locate ancestors back to a time $T$ we feel is reasonable (to the most recent glaciation) and use the dispersal estimate from that $T$.

Branching Brownian motion, also known as the Brownian–Yule process, is a simple model of spatial genealogies in a continuous population (*Malécot, 1948*; *Wright, 1943*). This simplicity comes at a cost: the lack of local density dependence generates non-uniform population density (*Felsenstein, 1975*) and the assumption of unbounded space overestimates the distance between samples (*Kalkauskas et al., 2021*). More complex models, such as the spatial Lambda Fleming–Viot process (*Barton et al., 2010*), overcome these issues and more, but are computationally expensive for inference (*Wirtz and Guindon, 2024*). Branching Brownian motion remains an analytically tractable model and a reasonable approximation over short-time scales (*Edwards, 1970*; *Rannala and Yang, 1996*; *Meligkotsidou and Fearnhead, 2007*; *Novembre and Slatkin, 2009*).

## Simulations

We first wanted to test the performance of our method in a situation where the true answers were known. To do this, we used a combination of spatially explicit forward-time simulations (*Haller and Messer, 2023*), coalescent simulations (*Kelleher et al., 2016a*), and tree-sequence tools (*Haller et al., 2019*; *Kelleher et al., 2019*; *Speidel et al., 2019*) to compare our estimates of dispersal rates and ancestor locations with the truth (see Materials and methods). This was also an opportunity to compare our estimates using the true trees vs. the Relate-inferred trees, to examine the influence of uncertainty in tree inference.

### Dispersal rates

Our method systematically underestimates simulated dispersal rates (*Figure 2A*), as expected given that our simple unbounded Brownian motion model allows the samples to be more broadly distributed than the finite habitat (e.g., *Ianni-Ravn et al., 2023*; *Kalkauskas et al., 2021*). Ignoring the distant past tends to reduce this underestimate (see $T = 1000$ in *Figure 2A*), but ignoring too much of the past increases noise and can lead to even larger underestimates (see $T = 100$ in *Figure 2A*). Despite the fact that we tend to underestimate the simulated dispersal rate, our estimates are highly correlated with the simulated values and we can interpret the dispersal rate inferred from the true trees as a true 'effective' dispersal rate given the habitat boundaries, local competition, etc.

Encouragingly, the dispersal estimates from the inferred trees are highly correlated with those from the true trees, with a slight upward bias. This upward bias can be explained by isolation-by-distance and errors in inferred tree topologies, causing geographically more distant samples to be mistakenly inferred as closer relatives. The upward bias may also result from Relate tending to underestimate longer coalescence times (*Y C Brandt et al., 2022*), which is consistent with the bias decreasing at smaller cutoff times. The bias increases as we use sample fewer trees at each locus (*Figure 2—figure supplement 1*), showing that our spatial prior implemented via importance sampling improves our inference. Sampling more individuals reduces noise across replicates but can actually increase bias (*Figure 2—figure supplement 2*), perhaps because of a greater chance of an error when inferring a larger tree. More work is needed to determine biases in ancestral recombination graphs inferred from spatial data more generally and to find additional ways to overcome this bias, for example, using only the most informative trees.

### Locating ancestors

We next wanted to test our ability to locate the genetic ancestor of a sampled genome at a given locus and a given time. With a single tree, our likelihood-based method gives both a point estimate (maximum likelihood estimate, MLE) and a 95% confidence ellipse (under the Brownian motion model), with only the latter relying on a genome-wide dispersal rate. With more than one tree we importance sample over likelihoods, numerically finding the maximum, which depends on the dispersal rate. We also have developed a best linear unbiased predictor (BLUP) of ancestral locations – importance sampling over analytically calculated MLE locations – that is faster to calculate, makes fewer assumptions, and is independent of the dispersal rate. Here, we focus on the full importance-sampled likelihood method for estimating ancestral locations, but the BLUP method is also implemented in our software and performs very similarly (*Figure 2—figure supplement 3*).

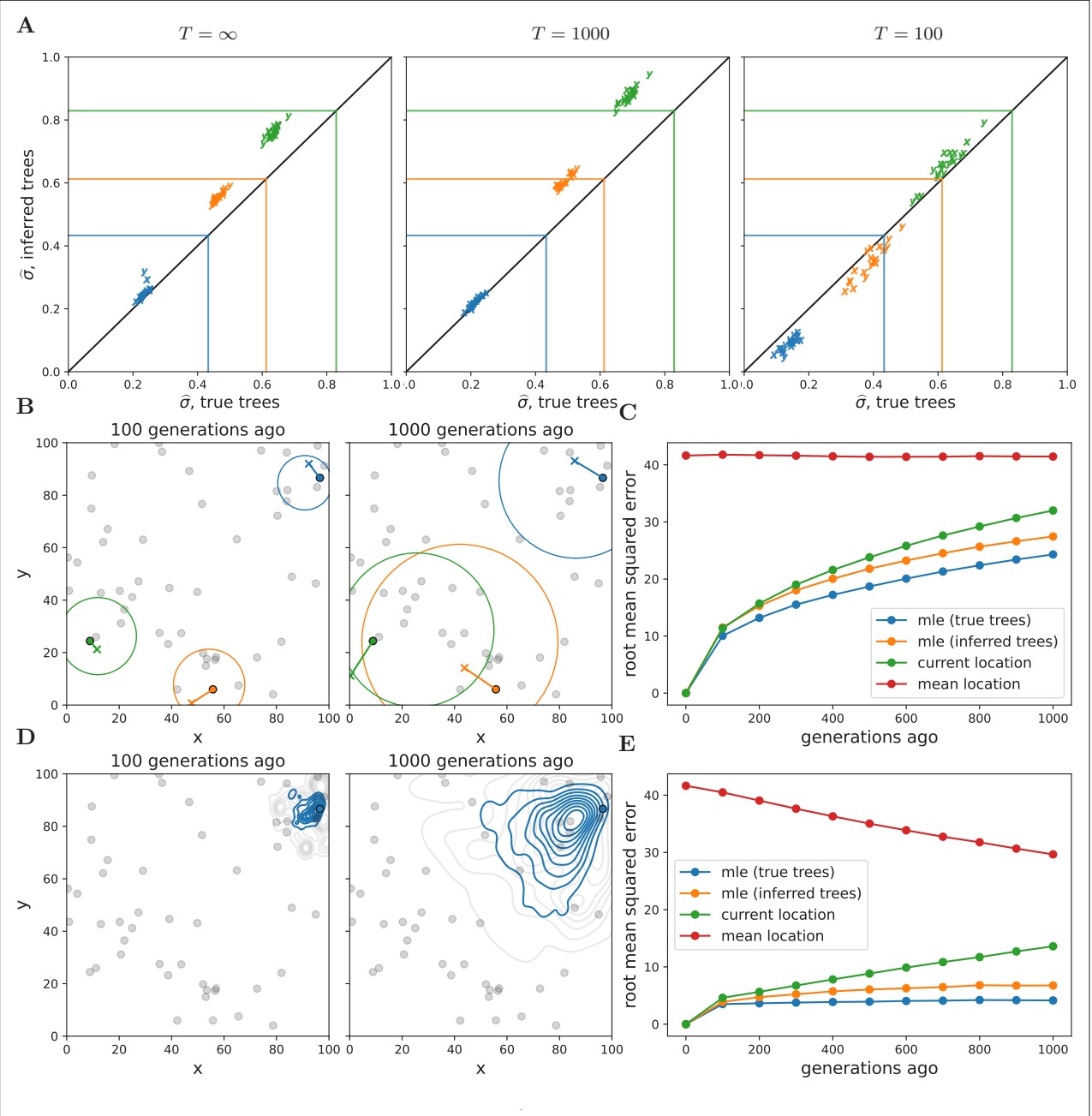

**Figure 2.** Simulations. (**A**) Accuracy of genome-wide dispersal rates. Maximum composite likelihood estimates of dispersal rate (in 'x' and 'y' dimensions) using the true trees vs. Relate-inferred trees for three different time cutoffs, $T$. Colours indicate the simulated dispersal rates, which are given by the corresponding lines. (**B**) Locating genetic ancestors at a particular locus. 95% confidence ellipses for the locations of genetic ancestors for three samples at a single locus (using the true trees and the simulated dispersal rate). The 'o's are the sample locations and the 'x's are the true ancestral locations. (**C**) Accuracy of locating genetic ancestors at individual loci. Root mean squared errors between the true locations of ancestors and the mean location of the samples (red), the current locations of the samples (green), and the maximum likelihood estimates from the inferred (orange) and true (blue) trees. (**D**) Locating genetic ancestors at many loci. Contour plots of the most likely (using the true trees; blue) and the true (grey) locations of genetic ancestors at every 100th locus for a given sample. (**E**) Accuracy of mean genetic ancestor locations. Root mean squared errors between the true mean location of genetic ancestors and the mean location of the samples (red), the current locations of the samples (green), and the mean maximum likelihood estimates from the inferred (orange) and true (blue) trees. To reduce computation time we only attempt to locate the first 10 samples. In all panels, there are 10 replicate simulations for each combination of time cutoff and dispersal rate. We sample 50 diploid individuals at random and use every 100th locus, with 1000 importance samples at each. Panels B–E have no time cutoff, $T = \infty$, and were simulated with a dispersal rate given by green lines in panel A. In panels C and E, the inferred tree ancestor location estimates use the inferred tree dispersal estimates.

The online version of this article includes the following figure supplement(s) for figure 2:

*Figure 2 continued on next page*

**Figure supplement 1.** The effect of importance sampling on dispersal estimates.

**Figure supplement 2.** The effect of sample size on dispersal estimates.

**Figure supplement 3.** Comparing maximum likelihood estimate (MLE) and best linear unbiased predictor (BLUP) ancestor locations.

**Figure supplement 4.** The effect of importance sampling on ancestor location estimates.

**Figure supplement 5.** The effect of sample size on ancestor location estimates.

*Figure 2B* shows the 95% confidence ellipses for the locations of the ancestors of three samples at one particular locus at two different times in the past, using the true trees and the simulated dispersal rate. While the ellipses generally do a good job of capturing true ancestral locations (the 'x's), the size of an ellipse at any one locus grows relatively rapidly as we move back in time, meaning that at deeper times (or higher dispersal rates) any one locus contains little information about an ancestor's location (as is the case for ancestral state reconstruction in phylogenetics; *Schluter et al., 1997*).

*Figure 2C* shows the resulting error in the MLE ancestor locations, using the true or inferred trees, and compares this to sensible straw-man estimates (the current location of each sample and the mean location across samples). While our method outperforms the straw-man estimates, the mean squared error in our inferred location of the ancestor at a locus grows relatively rapidly back in time, as expected under Brownian motion.

Given the large uncertainty of an ancestor's location at any one locus, we combine information across loci and consider a cloud of MLE ancestor locations from loci across the genome for a particular sample at a particular time in the past (*Figure 2D*). The genome-wide mean of the MLE locations remains closer to the true mean location of genetic ancestors (*Figure 2E*) than at any one locus, even with the inferred trees, suggesting our method can successfully trace major trends in the geographic ancestry of a sample deeper into the past. In contrast to dispersal, the error in ancestral locations appears relatively robust to the number of trees sampled per locus (*Figure 2—figure supplement 4*) and the number of samples (*Figure 2—figure supplement 5*). Importance sampling is expected to have a smaller effect on locations than dispersal because the locations depend much less on branch lengths. As we increase the number of samples we increase the expected number of topological errors but we also sample more of an ancestor's descendants and their close relatives, and so perhaps these two effects roughly cancel out.

## Empirical application: *A. thaliana*

*A. thaliana* has a complex, and not yet fully resolved, spatial history (*Fulgione and Hancock, 2018*; *Hsu et al., 2019*), including range expansions, admixture between multiple glacial refugia, and long-distance colonization. To further examine this history, we originally applied our method to 1135 *A. thaliana* accessions from a wide geographic range (*1001 Genomes Consortium, 2016*). Unfortunately, the dispersal rates we estimated from this data were unreasonably large, on the order of $10^4$ km²/generation, even after removing pairs of near-identical samples and geographic outliers (for more details see our preprint, *Osmond and Coop, 2021*). We next analysed an even larger dataset of roughly 1500 individuals from a broader geographic extent (*Durvasula et al., 2017*), but estimated a similarly large dispersal rate (results not shown but full pipeline available at https://github.com/mmosmond/spacetrees-ms, copy archived at *Osmond, 2024a*). The very high dispersal rate estimates from these datasets is an artefact of long runs of identity between pairs of geographically separated sequences (see Figure S6 of *Osmond and Coop, 2021*). We believe that these stretches of identity result from the incorrect imputation of missing genotypes, a problem that could not be resolved through various data filters. Imputation is required to infer the genealogies as Relate does not allow sites with missing genotypes. However, imputation can obscure rare alleles, which artificially lowers the divergence between similar sequences. This lowers coalescence times and biases dispersal estimates upward. We resolved this issue by analysing a smaller dataset with 66 long-read genomes (*Wlodzimierz et al., 2023*), where we could avoid imputing and capture more genetic variation (*Igolkina et al., 2024*) and therefore be much more confident in the resulting genealogies.

## *A. thaliana* genealogies

From 66 haploid genomes we used Relate (*Speidel et al., 2019*) to infer the genome-wide genealogy (87,518 trees), estimate effective population sizes through time (*Figure 3—figure supplement 1*), and resample branch lengths for importance sampling (see Materials and methods). The genealogies are publicly available at https://doi.org/10.5281/zenodo.11456353, which we hope will facilitate additional analyses (e.g., inferring selection; *Stern et al., 2019*; *Stern et al., 2021*).

A. thaliana is a selfer with a relatively low rate of outcrossing (*Bomblies et al., 2010*; *Platt et al., 2010*), thus it is worth taking a moment to consider the impact of selfing on our inferences. We chose A. thaliana because of its large sample size, broad geographic sampling, and intriguing spatial history. Furthermore, the availability of inbred accessions means that many samples have been well-studied and phasing is relatively straightforward. On the other hand, the high rate of selfing lowers the effective recombination rate and so is expected to increase the correlation in genealogies along the genome (*Nordborg, 2000*). However, in practice, linkage disequilibrium breaks down relatively rapidly in A. thaliana, on the scales of tens of kilobases (*Kim et al., 2007*), such that many trees along the genome should have relatively independent genealogies. A related issue is that the individuals with recent inbreeding (selfing) in their family tree will have fewer genealogical and genetic ancestors than outbred individuals. Thus in any recent time-slice there are a reduced number of independent genetic ancestors of a individual from a selfing population, but even with relatively low rates of outbreeding the number of ancestors still grows rapidly (*Lachance, 2009*), meaning that many trees along the genome should have relatively independent spatial histories. Finally, while the effective recombination rate may vary through time along with rates of selfing, Relate uses a mutational clock to estimate branch lengths, and thus they should be well calibrated to a generational time scale.

In our analyses below, we use every 100th tree starting from the beginning of each autosome, for a total of 878 trees (hereafter, loci). At each locus we resample the branch lengths 1000 times, giving us 1000 importance samples (hereafter, trees) at each locus.

## Estimating dispersal

We first used the trees at all sampled loci to estimate dispersal rate (*Figure 3*, inset). Using the full trees, back to the most recent common ancestor of each, we estimate a dispersal rate of about 30 km$^2$/generation.

To check that this dispersal estimate is well calibrated we can compare it to simpler estimates. To do this, we first calculated twice the average pairwise coalescent time over all loci (using just 1 tree per locus) for each pair of samples, $t_{ij}$ (multiplying $t_{ij}$ by the mutation rate then gives the expected pairwise nucleotide diversity; *Ralph et al., 2020*). A simple estimate of the dispersal rate is then the slope of the regression of the squared pairwise geographic distances, $d_{ij}^2$, on the average coalescent times, $t_{ij}$. This gives a dispersal estimate of roughly 10 km$^2$/generation, as does taking the average of $d_{ij}^2/t_{ij}$ over pairs (*Ianni-Ravn et al., 2023*). Replacing the pairwise coalescent times, $t_{ij}$, with pairwise nucleotide diversity divided by the mutation rate ($\pi_{ij}/(7 \times 10^{-9})$) gives a near-identical estimate that is independent of our inferred trees. Our tree-sequence estimate is expected to be larger than these simpler estimates because taking the arithmetic mean coalescent times over trees will lessen the amplifying effect of those trees with shorter coalescent times. Using instead the harmonic mean pairwise coalescent time over trees, we estimate a dispersal rate of roughly 40 km$^2$/generation, similar to our estimate.

We next examined the effect of cutting off our inference deeper in time (*Figure 3*, inset). Cutting the trees off 10$^6$ generations ago has essentially no effect, as it removes few coalescent events. With cutoffs of 10$^5$ and 10$^4$, the estimated dispersal rate increases to about 35 and 60 km$^2$/generation, respectively, which we believe is because finite habitat boundaries lower the effective dispersal rate over longer time scales but also suggests faster rates of lineage movement post-glaciation, for example, due to a recent northward range expansion. More recent cutoffs leave very few coalescent events and hence little information. We cannot reliably apply the same cutoffs to the simpler pairwise dispersal estimates discussed above because there are few pairs of samples that have an average genetic distance below the cutoffs, highlighting the increased amount of information in our method.

These dispersal estimates are considerably higher than a recent estimate, using machine learning directly on genotype data, that was on the order of 1 km$^2$/generation (*Smith et al., 2023*). This previous study used a similar sample size but a much narrower geographic range of more closely

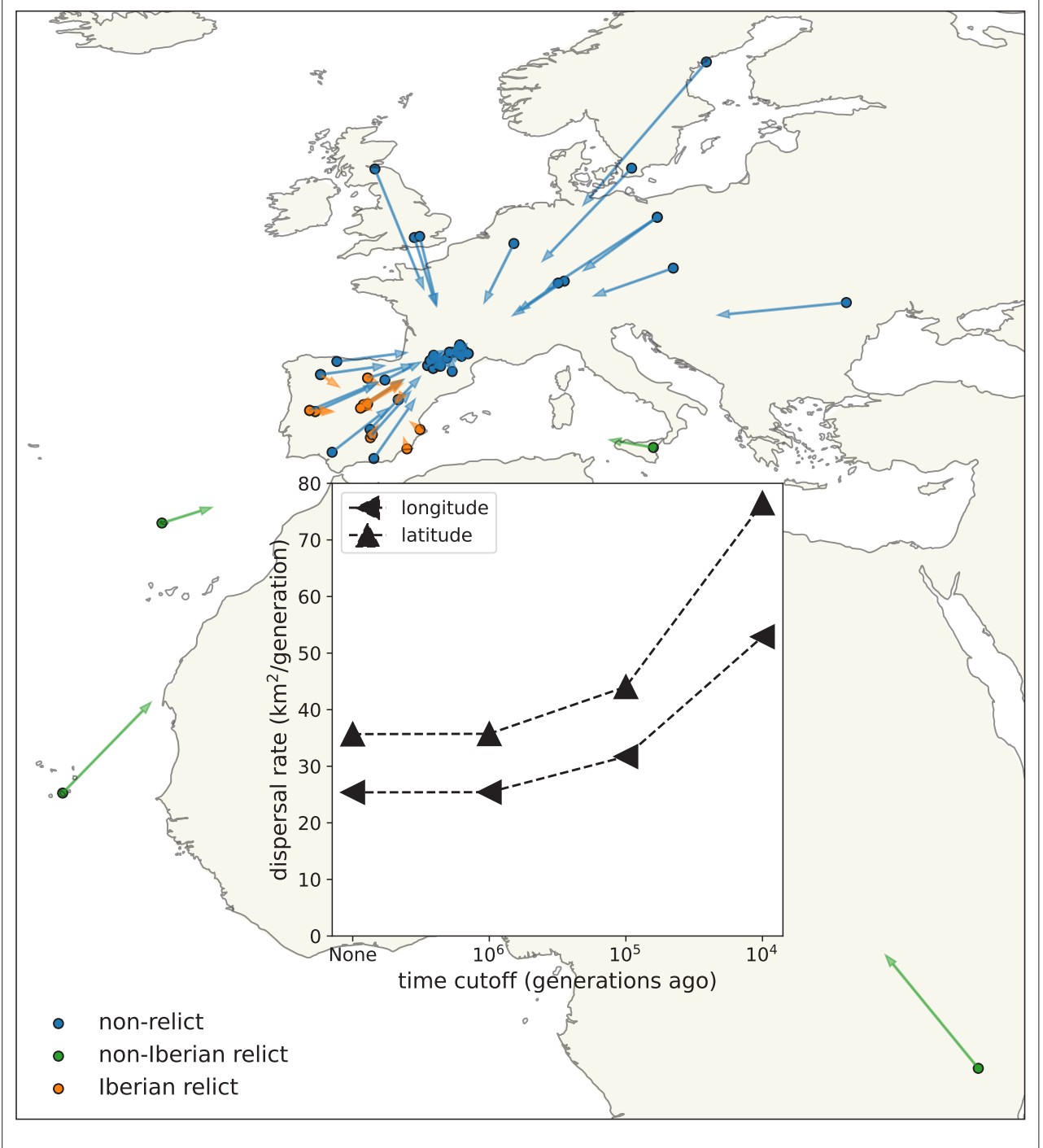

**Figure 3.** Dispersal rates (inset) and major trends in geographic ancestries. Vectors start at sample locations (circles) and point to the mean location of ancestors across 878 loci $10^4$ generations ago. Samples coloured by genomic principle component group (*Wlodzimierz et al., 2023*).

The online version of this article includes the following figure supplement(s) for figure 3:

**Figure supplement 1.** Relate-inferred effective population sizes and cross-coalescent rates.

related individuals. It is therefore likely that this lower estimate more strongly reflects shorter-term local dispersal, while we are striking a balance between these slower recent movements and faster movements that have occasionally occurred in deeper time. To get a sense of how large a dispersal rate we should expect, 'non-relict' lineages are proposed to have dispersed from the Black Sea to the

Atlantic Ocean (about 2500 km) in the past $10^4$ generations (*Lee et al., 2017*; *Hsu et al., 2019*). This suggests some lineages should have $d_i^2/t_i$ values of at least 300 km²/generation. We therefore think our effective dispersal rate is a reasonable one given the Brownian motion model and hypothesized movements in *A. thaliana*.

We estimate slightly faster dispersal along latitude than along longitude, especially in the past $10^4$ generations. This was confirmed by estimating the dispersal rate separately along latitude and longitude, removing dependence on the sample locations (see methods). Faster dispersal along latitude in the past $10^4$ generations is consistent with rapid post-glacial expansion. Previous studies have instead emphasized a rapid westward expansion of the non-relict lineages from an eastern glacial refugium, facilitated by relatively weak environmental gradients and, potentially, human movements and disturbance (*1001 Genomes Consortium, 2016*; *Lee et al., 2017*; *Hsu et al., 2019*). It is possible that our longitudinal dispersal estimate would increase substantially if we included non-relict samples from further east.

## Visualizing major trends in geographic ancestries

We next used the trees at all sampled loci and our dispersal estimate to locate genetic ancestors. Our method provides an estimate of the location of the ancestor at every locus for every sample back through time, giving a rich resource for visualizing the geography of genetic ancestry. As a first step, we visualize the mean ancestral location for every sample at a given time, averaged over loci, to visualize major geographic trends and detect samples with unusual geographic ancestries (*Figure 3*). When locating ancestors we use the full trees, which relate every sample to every other, but use the dispersal estimate from a time cutoff of $10^4$ generations and only locate ancestors back to that cutoff time.

Consistent with a rapid recent expansion of the non-relict ancestry (*Lee et al., 2017*; *Hsu et al., 2019*), lineages from the 'Eurasian' principle component group (hereafter, non-relicts) move relatively quickly towards each other when looking backwards in time, a visual display of their rapid coalescence over wide geographic distances. Looking forward through time (i.e., reversing the direction of the arrows), this rapid geographic expansion of non-relict ancestry is especially pronounced further north, as might be expected with retreating glaciers and more recent human disturbance (*Lee et al., 2017*). The fact that the non-relict lineages are all pulled towards the bulk of non-relict samples, in southern France, highlights the limitations of sampling and methods based on contemporary data alone. The non-relicts likely spread from further east, with a refugium perhaps near the Black Sea (*Lee et al., 2017*; *Hsu et al., 2019*), but this dataset does not have enough samples in that area to infer this.

Meanwhile nearly all lineages from the 'Iberian relict' principle component group (hereafter, Iberian relicts) move relatively little, and towards themselves rather than towards the non-relict ancestors. This is a visual representation of the fact that much of the ancestry of these Iberian relicts has a very different geographic history than much of the non-relicts' ancestry (*1001 Genomes Consortium, 2016*; *Fulgione and Hancock, 2018*). There are, however, two exceptions, with Iberian relicts Met-6 and Evs-12 having (overlapping) mean displacements that are much more typical of a non-relict. We explore these outliers more below.

Finally, we see that lineages from the 'non-Iberian relict' principle component group (hereafter, non-Iberian relicts) move towards the other samples, slightly more towards the Iberian relicts than the non-relicts. This movement is, however, relatively slow, emphasizing the known deep divergence of each of these samples with any other (*1001 Genomes Consortium, 2016*; *Fulgione et al., 2018*; *Fulgione and Hancock, 2018*; *Fulgione et al., 2022*).

## Visualizing the geographic sources of ancestry

We can also look at variation across loci in the geographic ancestry of a sample. We illustrate this while investigating the outliers above, the Iberian relicts with mean displacements more typical of a non-relict, Met-6 and Evs-12. In particular, we draw great circles connecting a sample's location to that of each of its ancestors at a given time, and include a histogram of the direction to the ancestors (a 'windrose' plot).

*Figure 4* compares four samples from Spain. The first is a typical non-relict (9883) whose ancestors are nearly all located to the northeast, as expected given the locations of the other non-relict samples and the proposed expansion from the east. The second is a typical Iberian relict (Hum-2) whose

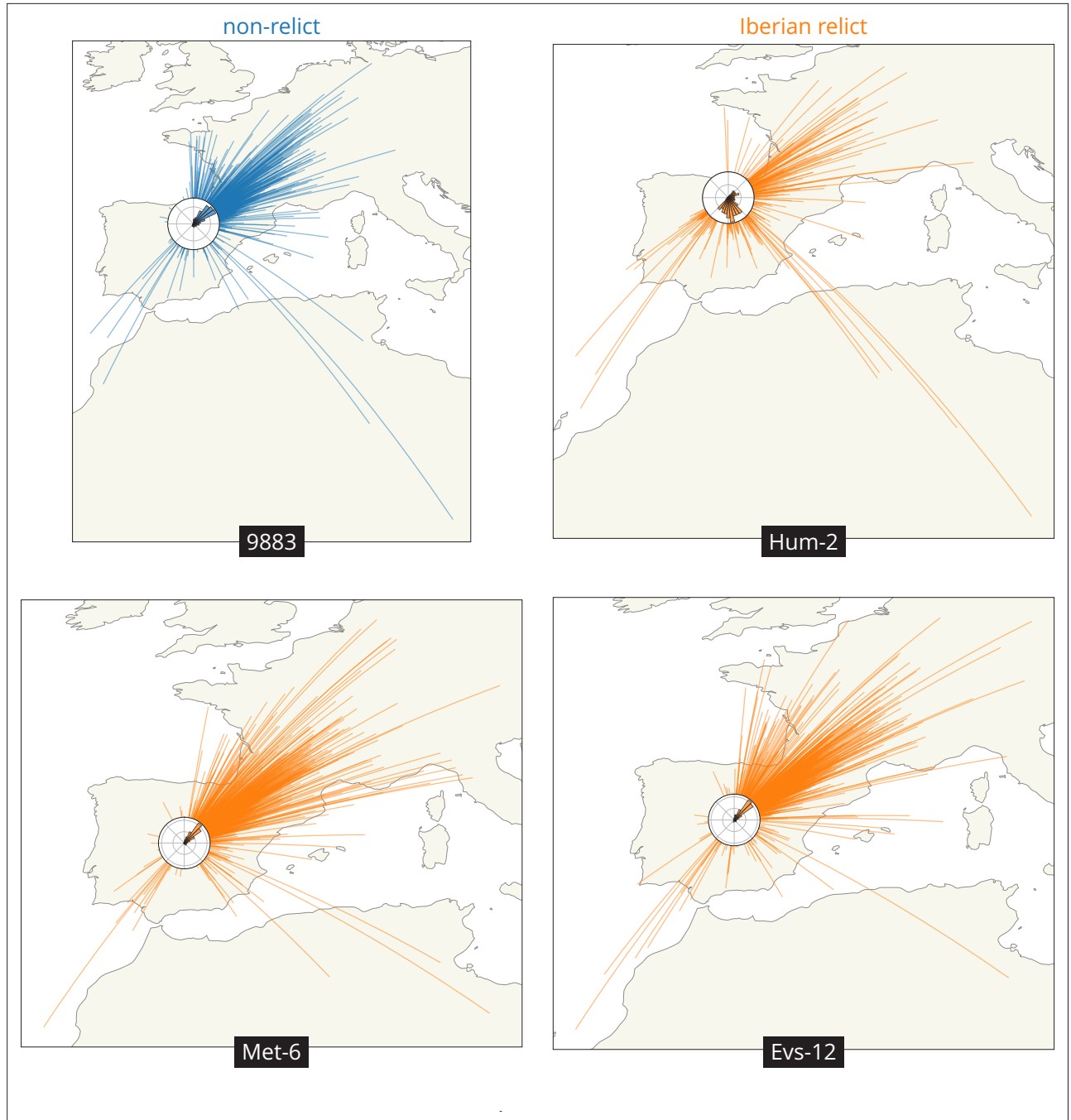

**Figure 4.** Visualizing the geographic sources of ancestry. Great circles connecting sample and ancestor locations at 878 loci 10⁴ generations ago. Polar histograms (windroses) show the distribution of direction in ancestral locations from the sample. Non-relicts in blue, Iberian relicts in orange.

ancestors are mostly located just to the south of it, near the centre of the Iberian relict samples, as expected given the divergence between non-relicts and relicts and the longer history of relicts in the region. This sample also has a substantial fraction of ancestors to the northeast, likely illustrating some recent admixture from the non-relict ancestry. The third and fourth samples are the outlier Iberian relicts (Met-6 and Evs-12), who both have many ancestors that are located relatively far to the north-east, much like the non-relict. This likely reflects quite extensive admixture between Iberian relicts and non-relicts (*1001 Genomes Consortium, 2016*; *Fulgione and Hancock, 2018*), which can also be seen in the fact that the genealogical nearest neighbour proportions (*Kelleher et al., 2019*) for these

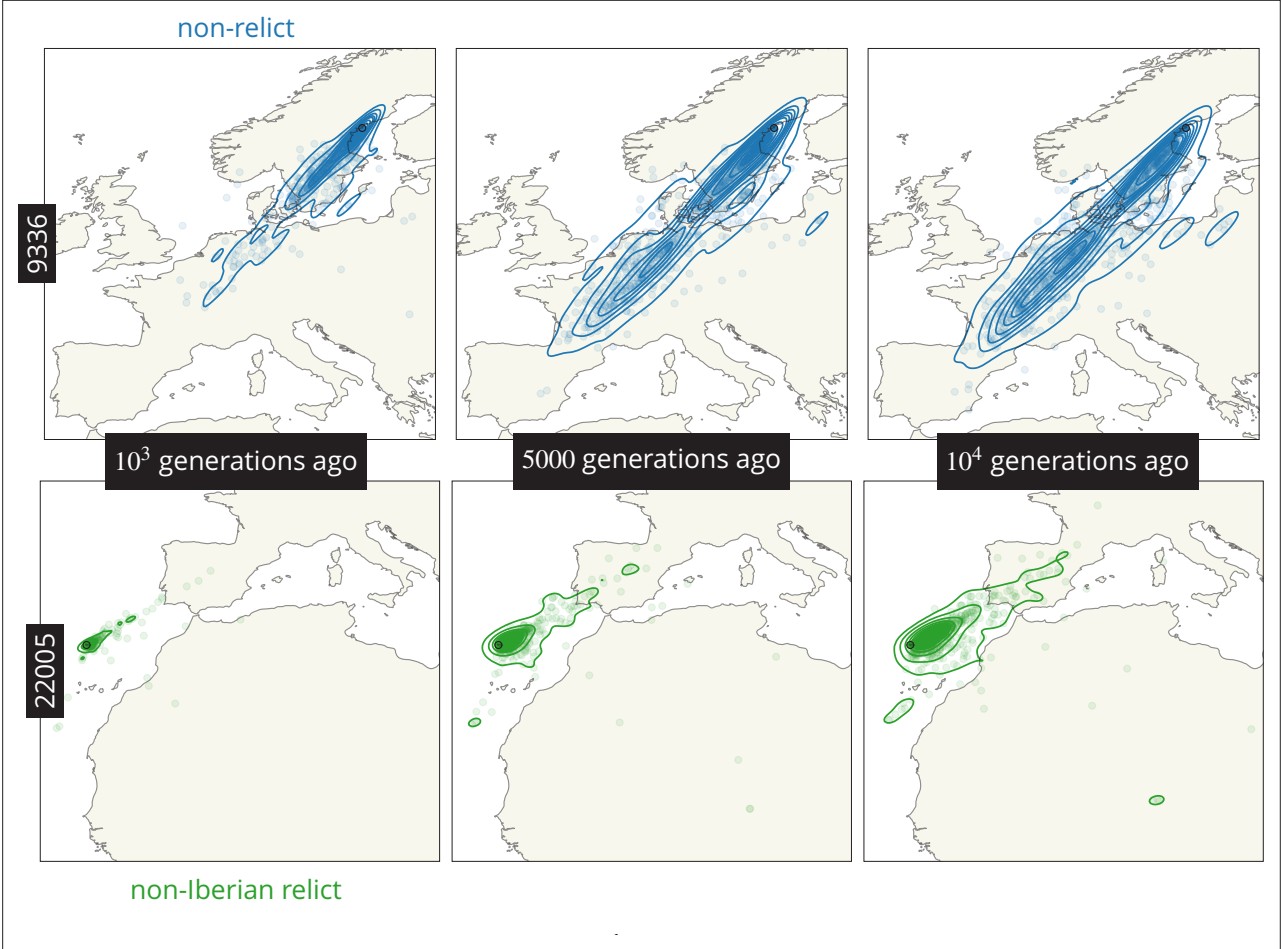

**Figure 5.** Visualizing the movement of geographic ancestries over time. Ancestor locations at 878 loci (circles) and kernel density estimates (contours). Non-relicts in blue, non-Iberian relcits in green.

two samples are heavily weighted towards non-relicts (at roughly 75% and 83%, while the mean across all Iberian relicts is roughly 30%).

## Visualizing the movement of geographic ancestries over time

We next looked at the entire distribution of genetic ancestor locations, over loci, for a given sample at a few given times (**Figure 5**). We see that the ancestors of the most northern sample (non-relict 9336) quickly move southwest towards the other non-relicts, illustrating a rapid northward expansion post-glaciation. In stark comparison, the ancestors of the most diverged sample (non-Iberian relict 22,005, in Madeira) move relatively little in the past $10^4$ generations, consistent with estimates that *A. thalaina* colonized Madeira roughly 70,000 years ago (**Fulgione and Hancock, 2018**).

There is a slight bimodality in the locations of the ancestors of the northern non-relict sample (9336) at later times. This bimodality also exists in the locations of the ancestors of the other sample from Sweden (non-relict 6137, not shown). It is tempting to wonder if this bimodality is a visualization of multiple pulses of northern expansions. In particular, based on the presence of abnormal amounts of relict ancestry in some northern samples, it has been hypothesized that relict lineages were present across much of Europe before being largely replaced, especially at midlatitudes, by a westward expansion of non-relict lineages (**Lee et al., 2017**; **Hsu et al., 2019**). These non-relict lineages are associated more with disturbed areas and so are expected to have arrived in the north more recently, and in lower numbers, explaining why some relict ancestry remains there. We therefore checked to see if the bimodality in ancestor locations of the Swedish samples could be explained by the placement of these focal samples in the corresponding trees. There was, however, no clear correlation between ancestor

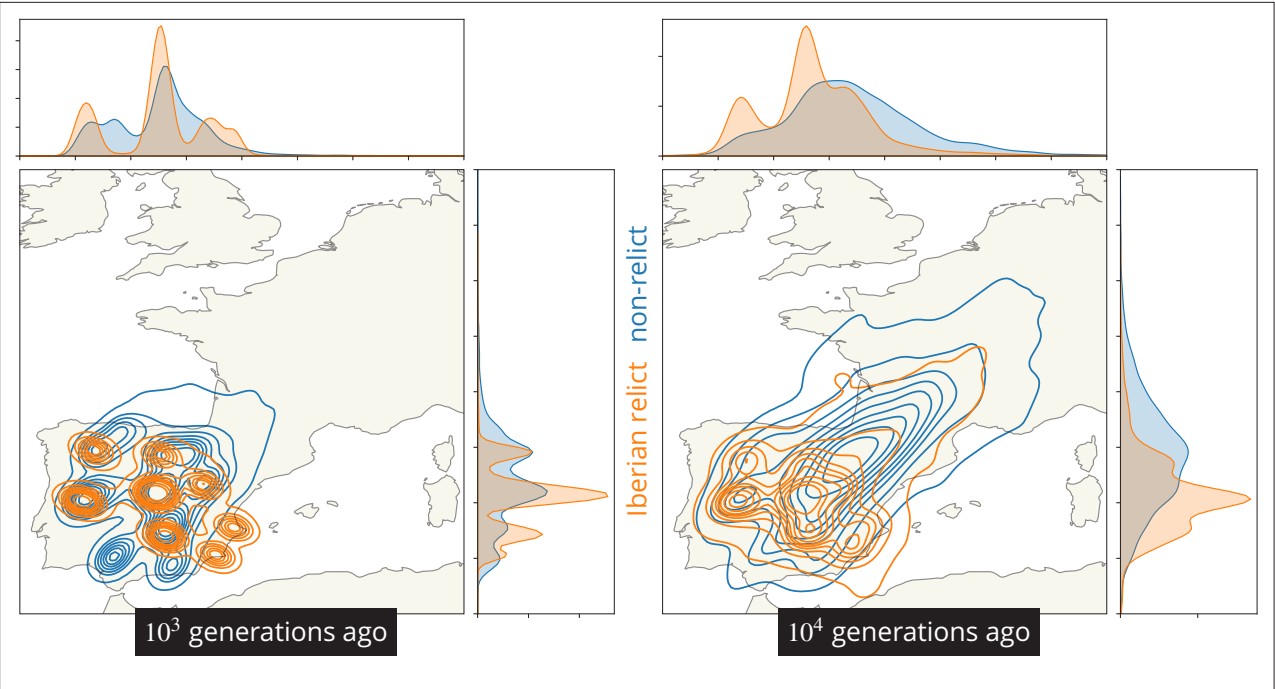

**Figure 6.** Visualizing the geographic ancestries of groups of samples. Kernel density estimates (contours and marginal distributions) of the locations of ancestors at 878 loci for all Iberian relicts (orange) and all non-relicts in Spain (blue).

location and the proportion of genealogical nearest neighbours that were non-relict vs. Iberian relict. It of course remains possible there were multiple bouts of colonization, perhaps even from within a given principle component group, that could be visualized with other metrics.

## Visualizing the geographic ancestries of groups of samples

The dataset contains samples drawing ancestors from at least two hypothesized glacial refugia, including the non-relcits with ancestry predominately from a refuge near the Black Sea (*Lee et al., 2017*; *Hsu et al., 2019*) and the Iberian relict samples with substantial ancestry from a refuge in northern Africa (*1001 Genomes Consortium, 2016*; *Durvasula et al., 2017*; *Fulgione and Hancock, 2018*). To visualize these alternative geographic ancestries we located the ancestors of all Iberian relicts and compared that distribution to the distribution of the locations of the ancestors of all non-relicts in mainland Spain (*Figure 6*). We see that the two ancestries are largely geographically overlapping in the past $10^3$ generations but then begin to diverge due to the majority of non-relicts ancestors moving northeast. There is also considerable variation in the locations of both ancestries, emphasizing substantial admixture. Having a few samples from north Africa would likely pull many of the ancestors of the Iberian relict samples south, further emphasizing the divergent geographic histories.

## Discussion

### Summary of main results

We have developed a method that uses a sequence of trees along a recombining genome – a genome-wide genealogy – to estimate individual-level dispersal rates and locate genetic ancestors (*Figure 1*). At the core of our method is a simple model of Brownian motion, allowing likelihoods to be quickly calculated from shared evolutionary times and sample locations. This also allows us to work in continuous space, negating the need to group individuals into discrete populations. On top of this we layer on importance sampling to correct for bias in inferred branch lengths and add a time cutoff to ignore strong violations of the model in the deep past. Simulation tests show that our method can estimate a meaningful effective dispersal rate and visualize major trends in geographic ancestries hundreds of generations into the past (*Figure 2*). Applying our method to *A. thaliana*, we estimate relatively rapid

dispersal since the last glaciation, especially across latitude (*Figure 3*, inset), and show how locating genetic ancestors allows us to visualize (1) major trends in geographic ancestries (*Figure 3*), (2) the geographic sources of ancestry (*Figure 4*), (3) movements of individual geographic ancestries over time (*Figure 5*), and (4) the geographic ancestries of groups of samples (*Figure 6*).

## Comparison with previous methods

The idea of using trees to estimate continuous ancestral characters and their rate of change is an old one. This was originally applied to population-level characters, such as the frequency of genes in a population and their rate of genetic drift, in a phylogenetic context (*Cavalli-sforza et al., 1964*; *Cavalli-Sforza and Edwards, 1967*; *Edwards, 1970*; *Felsenstein, 1973*; *Felsenstein, 1985*; *Grafen, 1989*). DNA sequencing later allowed the inference of a single tree relating individual samples for a sufficiently long non-recombining sequence (as found on the human Y chromosome, in a mito-chondrial genome, or in the nuclear genome of a predominately non-recombining species), which led to estimates of dispersal rates and ancestral locations under the banner of 'phylogeography' (*Avise, 2009*; *Knowles, 2009*). Phylogeography has proven incredibly useful, especially to infer the geographic origin and spread of viruses (*Biek et al., 2007*; *Lemey et al., 2009*; *Lemey et al., 2010*; *Bedford et al., 2010*; *Volz et al., 2013*), such as SARS-CoV-2 (*Worobey et al., 2020*; *Lemey et al., 2020*; *Dellicour et al., 2021a*; *Dellicour et al., 2021b*; *Martin et al., 2021*). Extending phylogeography to frequently recombining sequences is not straightforward as there are then many true trees that relate the samples. Only recently has it become feasible to infer the sequence of trees, and their branch lengths, along a recombining genome (*Rasmussen et al., 2014*; *Speidel et al., 2019*; *Wohns et al., 2021*). Our method capitalizes on this advance to use some of the enormous amount of information contained in a tree sequence of a large sample in a recombining species.

A related method was demonstrated by *Wohns et al., 2021*, who inferred the geographic location of coalescent nodes in a 'succinct' tree sequence (*Kelleher et al., 2018*), where information about nodes and edges are shared between trees. Our approach differs from theirs in a number of ways. First, they utilize the sharing of information about nodes and edges across trees to very efficiently geographically locate every node exactly once, placing each 'parent' node at the midpoint between its two 'child' nodes locations and iterating up the entire tree sequence simultaneously (rather than up each local tree individually). In addition to being fast, this has the advantage of using information from all the trees in a tree sequence. In contrast, we locate ancestors independently at each local tree we consider. As some ancestors (represented as nodes and edges) are shared between nearby trees along the genome (though we do not know precisely for how long since we lose that information when converting the Relate-inferred genealogy into a tree sequence), we avoid locating the same ancestors multiple times by sparsely sampling the trees (e.g., here we used every 100th tree). (Note that while we choose trees that have low linkage disequilibrium with one another and are therefore essentially unlinked, in the very recent past they will share ancestors but will quickly become independent; *Wakeley et al., 2012*.) Our approach therefore uses less of the information in the tree sequence, in this sense. On the other hand, when locating an ancestor we not only use information 'below' this ancestor (i.e., its descendants' locations and relations) but also the information 'above' the ancestor, due to the ancestor's lineage sharing time and a recent common ancestor with non-descendant lineages. Further advantages of our method include the ability to estimate dispersal rates and uncertainties in ancestor locations (as we have taken a para-metric approach), as well as accounting for uncertainty in branch lengths using importance sampling (this could be extended to capture uncertainty in the topologies as well once they can be efficiently sampled).

Ideally, we would merge the two methods, to efficiently use information from all the (correlated) trees in the tree sequence and all the information above and below ancestors. Two recent approaches have developed in this direction. One approach (*Grundler et al., 2024*) infers a dispersal rate and ancestral locations by placing each node at the location that minimizes a migration cost averaged over the trees in a succint tree sequence that the node appears in. A second approach (*Deraje et al., 2024*) extends the branching Brownian motion model to the full ancestral recombination graph (as inferred using a program like ARGweaver; *Rasmussen et al., 2014*), where the lineages meet each other forward-in-time at recombination nodes. Niether of these approaches yet account for uncertainty in tree inference.

## Future directions

We have chosen to use a very simple model of Brownian motion with a constant dispersal rate over time. Informally, we view this as a prior on the movement of the lineages, that can be overcome if the trees are informative about long-distance dispersal. However, there are a number of extensions that could readily be applied. For instance, we could allow dispersal rates to vary between branches (*O'Meara et al., 2006*) and compare dispersal rates in different parts of a species' range. Or we could model dispersal under a more complex model, like the early burst (*Harmon et al., 2010*) or a Lévy process (*Landis et al., 2013*), which may help identify periods of range expansion or sudden long-range dispersal. Alternatively, we could take a Bayesian approach, allowing much greater flexibility and the ability to incorporate many recent advances in phylogeography. For example, one could then model dispersal as a relaxed random walk (*Lemey et al., 2010*), which may be more appropriate for sample locations that are very non-normal, and could incorporate habitat boundaries. Or, given that there is large variance in the inferred locations of distant ancestors at any one locus (*Schluter et al., 1997*) but very many loci, we could take an 'empirical Bayes' approach and use the posteriors on ancestral locations over many loci to set a prior for a given locus. This might be especially helpful at deeper times, for example, tracing human ancestors back hundreds of thousands of years, where the noise in the ancestral location at any one locus is large, yet we can be relatively certain that the majority of lineages are in Africa. We might alternatively set priors to test hypotheses. For example, if we surmise there were multiple glacial refugia during the last glaciation we can set a prior on ancestral location with peaks at these hypothesized locations and infer what percent of a sample's lineages descended from each. Models of ancestral locations based on past climatic- and ecological-niche models could provide a rich source of data for building such priors and, given the large amounts data available in recombining sequences, these models could be subject to rigorous model choice. Finally, it might also be interesting to use machine learning. For example, disperseNN (*Smith et al., 2023*) and Locator (*Battey et al., 2020*) use neural networks to estimate dispersal rates and infer the locations of extant individuals, respectively, from unphased genotype data. In essence, this means these methods simultaneously determine the relationships between samples and estimate spatial parameters. Separating these two steps by first inferring a tree sequence and then supplying the structure of this tree sequence to a machine learning method (*Whitehouse et al., 2024*) may improve parameter estimates.

Our approach relies on the locations of the current day samples. While we have shown that we can learn much about the geographical history of a species with this approach, its accuracy is necessarily limited. For example, if historical parts of the range are undersampled in a particular dataset the method will struggle to locate ancestors in these regions, particularly further into the past, as we saw with the Iberian relicit samples with ancestry from a putative north African refugia. Similarly, if a species' range has shifted such that few present day individuals exist in portions of the historic range, as is likely the case of *A. thaliana* in central Africa (*Fulgione and Hancock, 2018*), we will often not infer ancestors to be in the currently sparsely occupied portions. Other large-scale movements, such as one ancestry replacing another (e.g., the non-relicts replacing the relicts in much of western Europe), may also partially obscure the geographic locations of ancestors. Over the past decade, we have learned about numerous large-scale movements in humans alongside the expansions of archaeological cultures, a fact fairly hidden from view by contemporary samples that only ancient DNA could bring to light (*Slatkin and Racimo, 2016*; *Reich, 2018*). One obvious way to improve our method then, is to include ancient samples. Given that it is now possible to include high-coverage ancient genomes in tree sequences (*Speidel et al., 2021*; *Wohns et al., 2021*), it is straightforward to include this information in our likelihoods (ancient samples are treated as any other, we calculate the shared times of these lineages with themselves and with all other sample lineages), influencing both our dispersal estimates and inferred ancestral locations. This should help, in particular, in locating ancestors that are closely related to the ancient samples and for detecting large-scale movements, such as range expansions, contractions, and replacements.

## Materials and methods

Here, we describe our methods to estimate dispersal rates and locate genetic ancestors and how we applied these to simulations and *A. thaliana*.

## Dispersal rate

### The probability distribution of the sample locations given a tree and the dispersal rate

Let $\mathbf{L}$ be the $n \times d$ matrix of sample locations, with $n$ samples in $d$ dimensions. We first derive the probability distribution of $\mathbf{L}$ given a tree topology (a directed graph with $n-1$ nodes and $2n-2$ edges), $\mathcal{G}$, and associated branch lengths (a vector of $2n-2$ times), $\mathcal{T}$, which together describe the coalescent history of the sample, and the dispersal rate ($d \times d$ covariance matrix), $\boldsymbol{\Sigma}$. We derive this probability distribution, $\mathbb{P}(\mathbf{L}|\mathcal{G},\mathcal{T},\boldsymbol{\Sigma})$, by compounding the normally distributed dispersal events each generation to give Brownian motion down the tree in a similar manner to phylogenetic least squares regression (*Grafen, 1989*; *Cornwell and Nakagawa, 2017*).

Let $\mathbf{S}_{\mathcal{G},\mathcal{T}}$ be the $n \times n$ matrix of shared times (in generations) between each pair of sample lineages back to the root in the tree defined by $\mathcal{G}$ and $\mathcal{T}$. Then, assuming per generation dispersal is multivariate normal with mean displacement 0 and covariance matrix $\boldsymbol{\Sigma}$, the probability distribution of the locations depends only on the shared times in the tree, $\mathbb{P}(\mathbf{L}|\mathcal{G},\mathcal{T},\boldsymbol{\Sigma}) = \mathbb{P}(\mathbf{L}|\mathbf{S}_{\mathcal{G},\mathcal{T}},\boldsymbol{\Sigma})$. In fact, the locations are normally distributed around the MLE for the location of the most recent common ancestor, $\widehat{\ell}_A = [(\mathbf{1}^{\mathsf{T}}\mathbf{S}_{\mathcal{G},\mathcal{T}}^{-1}\mathbf{1})^{-1}(\mathbf{1}^{\mathsf{T}}\mathbf{S}_{\mathcal{G},\mathcal{T}}^{-1}\mathbf{L})]$ with a covariance that is the (Kronecker) product of the shared times and the dispersal rate, $\mathbf{S}_{\mathcal{G},\mathcal{T}} \otimes \boldsymbol{\Sigma}$. Here, $\mathbf{1}$ is a column vector of $n$ ones.

### Mean centring

We can remove any dependence on the (unknown) location of the most recent common ancestor, $\ell_A$, by mean centring the data (e.g., *Lee and Coop, 2017*). In particular, we subtract the mean sample location from each sample location and drop one of the samples (as we lose a degree of freedom by centring). The mean-centred locations can be computed with $\mathbf{X} = \mathbf{ML}$, where $\mathbf{M}$ is an $(n-1) \times n$ matrix with $(n-1)/n$ on the diagonal and $-1/n$ elsewhere. The mean-centred shared times are $\mathbf{V}_{\mathcal{G},\mathcal{T}} = \mathbf{M}\mathbf{S}_{\mathcal{G},\mathcal{T}}\mathbf{M}^{\mathsf{T}}$.

Similar to above, the probability distribution of the mean-centred locations only depends on the shared times, $\mathbb{P}(\mathbf{X}|\mathcal{G},\mathcal{T},\boldsymbol{\Sigma}) = \mathbb{P}(\mathbf{X}|\mathbf{V}_{\mathcal{G},\mathcal{T}},\boldsymbol{\Sigma})$. These mean-centred locations are normally distributed around 0 with covariance $\mathbf{V}_{\mathcal{G},\mathcal{T}} \otimes \boldsymbol{\Sigma}$.

### Chopping the tree

We will usually want to only consider lineages more recently than some cutoff time, $T$, since deeper genealogical history may contain little geographic information (*Wilkins, 2004*). When this is the case we cut the full tree of at $T$, leaving us with $n_T \in [1,n]$ subtrees, where all coalescent times are $\leq T$. Treating each subtree independently, the probability distribution of the sample locations is the product of the probability distributions for the sample locations in each subtree,

$$\mathbb{P}(\mathbf{X}|\mathcal{G},\mathcal{T},\boldsymbol{\Sigma},T) = \prod_{i=1}^{n_T} \mathbb{P}(\mathbf{X}_i|\mathcal{G}_i,\mathcal{T}_i,\boldsymbol{\Sigma}) = \prod_{i=1}^{n_T} \mathbb{P}(\mathbf{X}_i|\mathbf{V}_{\mathcal{G}_i,\mathcal{T}_i},\boldsymbol{\Sigma}),$$

where $\mathcal{G}_i$ is the topology, $\mathcal{T}_i$ is the branch lengths, $\mathbf{V}_{\mathcal{G}_i,\mathcal{T}_i}$ is the (mean-centred) shared times, and $\mathbf{X}_i$ is the (mean-centred) sample locations for subtree $i$. The (mean-centred) locations in subtree $i$ are normally distributed around 0 with covariance $\mathbf{V}_{\mathcal{G}_i,\mathcal{T}_i} \otimes \boldsymbol{\Sigma}$.

Note that to mean centre we need at least two samples in a subtree. Given that we do not know the root location, when there is only one sample we do not have enough information to infer a dispersal rate from this subtree. We therefore ignore subtrees with only one sample.

### Importance sampling

The calculations above, which give the likelihood of the dispersal rate given the sample locations, are all predicated on knowing the tree with certainty, which will not be the case when inferring trees from genetic data. Furthermore, inferring a tree may involve assumptions (such as panmixia) that are inconsistent with the model we are using. To deal with uncertainty and sampling bias, we calculate the likelihood of our parameters given the data using importance sampling, a likelihood ratio, and Monte Carlo approximation. Importance sampling corrects the expectation of the likelihood by reweighting draws from an 'incorrect' (proposal) distribution to match the 'correct' (target) distribution. These

importance weights downweight draws from the proposal distribution that are likely under the proposal distribution but unlikely under the target distribution and upweight draws that are likely in the target distribution and unlikely under the proposal distribution. Importance sampling has been applied to genealogies in a number of population genetic settings as coalescent models provide convenient priors on trees and it is often challenging to sample genealogies consistent with data (*Griffiths and Tavare, 1997*; *Stephens and Donnelly, 2000*; *Meligkotsidou and Fearnhead, 2007*). See *Stern et al., 2019Stern et al., 2021*; *Stern et al., 2019* for recent applications of these ideas to marginal trees inferred along a recombining sequence, whose general approach we follow below.

The data we have are the locations of the samples, $\mathbf{L}$, and their haplotypes, $\mathbf{H}$ (a matrix with genomes as rows and sites as columns). From this data we want to infer two unknowns: a tree topology, $\mathcal{G}$, and the associated branch lengths, $\mathcal{T}$. We then want to use these unknowns to estimate the likelihood of the dispersal rate, $\mathbf{\Sigma}$. Put another way, we want to estimate $\mathbb{E}_{\mathcal{G},\mathcal{T}|\mathbf{\Sigma}}\left[\mathbb{P}(\mathbf{L},\mathbf{H}|\mathcal{G},\mathcal{T},\mathbf{\Sigma})\right]$, which is the likelihood of our parameters given the data, integrated over the unknowns.

Current methods to infer tree topologies and branch lengths along a recombining sequence (*Rasmussen et al., 2014*; *Speidel et al., 2019*; *Kelleher et al., 2019*; *Wohns et al., 2021*) assume panmictic, well-mixed populations. This implies we cannot sample topologies and branch lengths, $\mathcal{G}$ and $\mathcal{T}$, under our spatial model, creating bias. We correct this bias with importance sampling, weighting each likelihood by the probability of the topology and branch lengths under our spatial model, $\mathbb{P}(\mathcal{G},\mathcal{T}|\mathbf{\Sigma})$, relative to their probability under the panmictic model we are sampling from, $\mathbb{P}(\mathcal{G},\mathcal{T}|\mathbf{H},\text{panmixia})$. This allows us to integrate over topology and branch lengths from the panmictic model, giving a dispersal rate likelihood

$$
\begin{aligned}
\mathrm{L}(\mathbf{\Sigma}) &:= \mathbb{E}_{\mathcal{G},\mathcal{T}|\mathbf{\Sigma}}\left[\mathbb{P}(\mathbf{L},\mathbf{H}|\mathcal{G},\mathcal{T},\mathbf{\Sigma})\right] \\
&= \mathbb{E}_{\mathcal{G},\mathcal{T}|\mathbf{H},\text{panmixia}}\left[\frac{\mathbb{P}(\mathbf{L},\mathbf{H}|\mathcal{G},\mathcal{T},\mathbf{\Sigma})\mathbb{P}(\mathcal{G},\mathcal{T}|\mathbf{\Sigma})}{\mathbb{P}(\mathcal{G},\mathcal{T}|\mathbf{H},\text{panmixia})}\right].
\end{aligned}
\tag{1}
$$

The probabilities containing the genetic data, $\mathbf{H}$, are complicated to calculate. To simplify we divide by the likelihood of panmixia given the data, $\mathrm{L}(\text{panmixia}) = \mathbb{P}(\mathbf{H}|\text{panmixia})$, to work with the likelihood ratio

$$
\begin{aligned}
\mathrm{LR}(\mathbf{\Sigma}) &:= \frac{\mathrm{L}(\mathbf{\Sigma})}{\mathrm{L}(\text{panmixia})} \\
&= \mathbb{E}_{\mathcal{G},\mathcal{T}|\mathbf{H},\text{panmixia}}\left[\frac{\mathbb{P}(\mathbf{L},\mathbf{H}|\mathcal{G},\mathcal{T},\mathbf{\Sigma})\mathbb{P}(\mathcal{G},\mathcal{T}|\mathbf{\Sigma})}{\mathbb{P}(\mathcal{G},\mathcal{T}|\mathbf{H},\text{panmixia})\mathbb{P}(\mathbf{H}|\text{panmixia})}\right] \\
&= \mathbb{E}_{\mathcal{G},\mathcal{T}|\mathbf{H},\text{panmixia}}\left[\frac{\mathbb{P}(\mathbf{L},\mathbf{H}|\mathcal{G},\mathcal{T},\mathbf{\Sigma})\mathbb{P}(\mathcal{G},\mathcal{T}|\mathbf{\Sigma})}{\mathbb{P}(\mathbf{H},\mathcal{G},\mathcal{T}|\text{panmixia})}\right] \\
&= \mathbb{E}_{\mathcal{G},\mathcal{T}|\mathbf{H},\text{panmixia}}\left[\frac{\mathbb{P}(\mathbf{H}|\mathcal{G},\mathcal{T})\mathbb{P}(\mathbf{L}|\mathcal{G},\mathcal{T},\mathbf{\Sigma})\mathbb{P}(\mathcal{G},\mathcal{T}|\mathbf{\Sigma})}{\mathbb{P}(\mathbf{H}|\mathcal{G},\mathcal{T})\mathbb{P}(\mathcal{G},\mathcal{T}|\text{panmixia})}\right] \\
&= \mathbb{E}_{\mathcal{G},\mathcal{T}|\mathbf{H},\text{panmixia}}\left[\frac{\mathbb{P}(\mathbf{L}|\mathcal{G},\mathcal{T},\mathbf{\Sigma})\mathbb{P}(\mathcal{G},\mathcal{T}|\mathbf{\Sigma})}{\mathbb{P}(\mathcal{G},\mathcal{T}|\text{panmixia})}\right].
\end{aligned}
\tag{2}
$$

The third step assumes the genetic data ($\mathbf{H}$) is conditionally independent of the spatial parameters ($\mathbf{\Sigma}$ or panmixia) and locations ($\mathbf{L}$) given the tree ($\mathcal{G}$ and $\mathcal{T}$). We can approximate this expectation using Monte Carlo sampling,

$$
\widehat{\mathrm{LR}(\mathbf{\Sigma})} = \frac{1}{M}\sum_{m=1}^{M}\frac{\mathbb{P}(\mathbf{L}|\mathcal{G}_m,\mathcal{T}_m,\mathbf{\Sigma})\mathbb{P}(\mathcal{G}_m,\mathcal{T}_m|\mathbf{\Sigma})}{\mathbb{P}(\mathcal{G}_m,\mathcal{T}_m|\text{panmixia})},
\tag{3}
$$

where the genealogies and branch lengths, $\mathcal{G}_m$ and $\mathcal{T}_m$, are sampled from the panmictic model, $\mathbb{P}(\mathcal{G},\mathcal{T}|\mathbf{H},\text{panmixia})$, using Markov chain Monte Carlo.

We make two final simplifications. First, we will use a model of branching Brownian motion for $\mathbb{P}(\mathcal{G},\mathcal{T}|\mathbf{\Sigma})$ and the standard neutral coalescent for $\mathbb{P}(\mathcal{G},\mathcal{T}|\text{panmixia})$. Under both of these models the probability of the topology is equivalent (and uniform), $\mathbb{P}(\mathcal{G}|\mathbf{\Sigma}) = \mathbb{P}(\mathcal{G}|\text{panmixia})$. Second, we will use Relate (*Speidel et al., 2019*) to infer topologies and branch lengths. Relate returns a single topology and allows resampling over branch lengths conditional on this topology. We therefore take the

topology as given and integrate only over branch lengths. Putting these two simplifications together, the likelihood ratio becomes

$$
\begin{aligned}
\widehat{\mathrm{LR}(\boldsymbol{\Sigma})} &= \frac{1}{M} \sum_{m=1}^{M} \frac{\mathbb{P}(\mathbf{L}|\mathcal{G}_m, \mathcal{T}_m, \boldsymbol{\Sigma})\mathbb{P}(\mathcal{G}_m, \mathcal{T}_m|\boldsymbol{\Sigma})}{\mathbb{P}(\mathcal{G}_m, \mathcal{T}_m|\mathrm{panmixia})} \\
&= \frac{1}{M} \sum_{m=1}^{M} \frac{\mathbb{P}(\mathbf{L}|\mathcal{G}_m, \mathcal{T}_m, \boldsymbol{\Sigma})\mathbb{P}(\mathcal{T}_m|\mathcal{G}_m, \boldsymbol{\Sigma})\mathbb{P}(\mathcal{G}_m|\boldsymbol{\Sigma})}{\mathbb{P}(\mathcal{T}_m|\mathcal{G}_m, \mathrm{panmixia})\mathbb{P}(\mathcal{G}_m|\mathrm{panmixia})} \\
&= \frac{1}{M} \sum_{m=1}^{M} \frac{\mathbb{P}(\mathbf{L}|\mathcal{G}_m, \mathcal{T}_m, \boldsymbol{\Sigma})\mathbb{P}(\mathcal{T}_m|\mathcal{G}_m, \boldsymbol{\Sigma})}{\mathbb{P}(\mathcal{T}_m|\mathcal{G}_m, \mathrm{panmixia})} \\
&\approx \frac{1}{M} \sum_{m=1}^{M} \frac{\mathbb{P}(\mathbf{L}|\mathcal{G}, \mathcal{T}_m, \boldsymbol{\Sigma})\mathbb{P}(\mathcal{T}_m|\mathcal{G}, \boldsymbol{\Sigma})}{\mathbb{P}(\mathcal{T}_m|\mathcal{G}, \mathrm{panmixia})},
\end{aligned}
\tag{4}
$$

where the branch lengths, $\mathcal{T}_m$, are sampled from $\mathbb{P}(\mathcal{T}|\mathbf{H}, \mathrm{panmixia}, \mathcal{G})$ using Markov chain Monte Carlo. Note that because the probability of a topology is the same in the two models, and therefore cancels out, our method can immediately be extended to integrate over both topologies and branch lengths. Future work could integrate over both topologies and branch lengths with other software (*Rasmussen et al., 2014*; *Wohns et al., 2021*) or by running Relate on subsamples of the genomes and integrating over subsamples.

We next show how we derive the probability distributions of the branch lengths in the tree that we use as the weights in the importance sampler, $\mathbb{P}(\mathcal{T}|\mathcal{G}, \mathrm{panmixia})$ and $\mathbb{P}(\mathcal{T}|\mathcal{G}, \boldsymbol{\Sigma})$.

## The probability distribution of the branch lengths given a tree topology and panmixia

Relate and other tree construction methods infer the branch lengths in the trees under a model of panmixia with variable population size through time (with this demographic history inferred from the same haplotypes as part of the method). The probability distribution of these branch lengths given the tree topology and panmixia, $\mathbb{P}(\mathcal{T}|\mathcal{G}, \mathrm{panmixia})$, can be derived from the standard neutral coalescent allowing for (stepwise) changing effective population size (*Griffiths and Tavaré, 1994*; *Meligkotsidou and Fearnhead, 2007*).

Let the number of samples be $n$ and let $u_i$ be the $i^{\mathrm{th}}$ most recent coalescence time (as determined by the branch lengths). The coalescence times are produced by a Markov process, where the probability of the $i^{th}$ coalescence being at time $u_i$ is independent of the other coalescence times once conditioned on the time of the previous coalescence, $u_{i-1}$. The probability distribution of the branch lengths can therefore be written

$$
\mathbb{P}(\mathcal{T}|\mathcal{G}, \mathrm{panmixia}) = \prod_{i=1}^{n-1} \mathbb{P}(U_i = u_i|U_{i-1} = u_{i-1}).
\tag{5}
$$

Between time $u_{i-1}$ and $u_i$ we have $n - (i - 1)$ lineages. The coalescence time $u_i$ is then approximately distributed as a time-inhomogeneous exponential random variable with instantaneous rate $\lambda(u) = \binom{n-(i-1)}{2}/[2N(u)]$, where $N(t)$ is the effective population size at time $t$ in the past, giving

$$
\begin{aligned}
&\mathbb{P}(U_i = u_i|U_{i-1} = u_{i-1}) \\
&= \binom{n-(i-1)}{2} \frac{1}{2N(u_i)} \exp\left[-\binom{n-(i-1)}{2}[\Lambda(u_i) - \Lambda(u_{i-1})]\right]
\end{aligned}
\tag{6}
$$

with

$$
\Lambda(u) = \int_0^u \frac{1}{2N(t)}\mathrm{d}t
\tag{7}
$$

the cumulative coalescence rate up to time $u$.

When the effective population size is piecewise constant, split into $K$ epochs with effective population size $N_k$ between time $\tau_k$ and $\tau_{k+1}$, then the cumulative coalescence rate is (see Appendix A of **Stern et al., 2021**)

$$\Lambda(u) = \left[ \sum_{k=0}^{b(u)} \frac{1}{2N_k} (\tau_{k+1} - \tau_k) \right] + \frac{1}{2N_{b(u)+1}} (u - \tau_{b(u)+1}),$$
(8)

where $b(u) = \max(k \in \{0, 1, 2, ..., K-1\} : u > \tau_{k+1})$ is the last epoch to end before time $u$.

If we only want the probability density of coalescence times back as far as time $T$, we consider only these $m \leq n - 1$ coalescence times and then multiply the probability density of these times by the probability of no coalescence between time $u_m$ and $T$,

$$\mathbb{P}(\mathcal{T}|\mathcal{G}, \text{panmixia}, T)$$
$$= \exp\left[ -\binom{n-m}{2} [\Lambda(T) - \Lambda(u_m)] \right] \prod_{i=1}^{m} \mathbb{P}(U_i = u_i | U_{i-1} = u_{i-1}).$$
(9)

## The probability distribution of the branch lengths given a tree topology and our spatial model

A number of spatial models of the coalescent have been developed. However, for our purposes many are computational challenging to work with and so we pursue a simpler, analytically tractable approximation called branching Brownian motion, or the Brownian–Yule process. Such approximations have been used previously to approximate genealogies on short time scales (**Edwards, 1970**; **Rannala and Yang, 1996**; **Meligkotsidou and Fearnhead, 2007**). We view the rate of branching ($\lambda$) in the branching Brownian motion process as a nuisance parameter, but informally it is proportional to the inverse of the local population density and so it might be a useful parameter to explore in future work (for a recent application see **Ringbauer et al., 2017**).

Let $T$ be the time in the past we wish to start the process (which in practice will be no greater than the time to the most recent common ancestor in the tree). Let $n_0$ be the number of ancestral lineages at this time. Taking a forward in time perspective, let $u'_i$ be the time at which we go from $i$ to $i+1$ lineages ($u'_i = T - u_{n-i}$). Then under a pure birth (Yule) process with per capita birth rate $\lambda$ the joint probability density of the $n - n_0$ birth times and ending up with $n$ samples (i.e., no birth once $n$ lineages existed) is

$$\mathbb{P}(\mathbf{u}', n|\lambda, n_0, T) = \left( \prod_{i=n_0}^{n-1} \lambda i \exp[-\lambda i (u'_i - u'_{i-1})] \right) \exp[-\lambda n (T - u'_{n-1})]$$
$$= \lambda^{n-n_0} \frac{(n-1)!}{(n_0-1)!} \exp\left[ -\lambda \left( nT + \sum_{i=n_0}^{n-1} u'_i \right) \right],$$
(10)

where we have defined the starting time of the process $u'_{n_0-1} = 0$.

The probability that $n_0$ lineages produce exactly $n$ lineages in time $T$ under a pure birth process is given by a negative binomial with $n - n_0$ successes, $n_0$ failures, and success probability $1 - \exp(-\lambda T)$,

$$\mathbb{P}(n|\lambda, n_0, T) = \binom{n-1}{n-n_0} \exp(-\lambda T n_0)[1 - \exp(-\lambda T)]^{n-n_0}.$$
(11)

The probability distribution of the branch lengths under the Yule process, which we take to be the probability of the branch lengths given the topology and the spatial model, is therefore proportional to the ratio of **Equations 10 and 11**,

$$\mathbb{P}(T|G, \boldsymbol{\Sigma}) = \mathbb{P}(\mathbf{u}'|\lambda, n_0, T, n)$$

$$\propto \frac{\mathbb{P}(\mathbf{u}', n|\lambda, n_0, T)}{\mathbb{P}(n|\lambda, n_0, T)} \tag{12}$$

$$= (n - n_0)! \left( \frac{\lambda \exp(-\lambda T)}{1 - \exp(-\lambda T)} \right)^{n-n_0} \exp\left( -\lambda \sum_{i=n_0}^{n-1} u'_i \right).$$

Note that the branching times are independent of the dispersal rate given the topology, $\mathbb{P}(T|G, \boldsymbol{\Sigma}) = \mathbb{P}(T|G)$, under this relatively simple spatial model. See *Edwards, 1970*; *Rannala and Yang, 1996*; *Meligkotsidou and Fearnhead, 2007* for more on the Yule process.

## Conditioning on sampling locations

As many studies choose sampling locations before choosing samples, we would like to condition on sampling locations in our inferences of dispersal rate, i.e., use the conditional likelihood $\mathbb{P}(\mathbf{L}|\mathcal{G}, \mathcal{T}, \boldsymbol{\Sigma})/\mathbb{P}(\mathbf{L}|\boldsymbol{\Sigma})$. The denominator, $\mathbb{P}(\mathbf{L}|\boldsymbol{\Sigma})$, is the probability distribution of the locations of the descendent tips of a branching Brownian motion given the dispersal rate. This is the numerator, $\mathbb{P}(\mathbf{L}|\mathcal{G}, \mathcal{T}, \boldsymbol{\Sigma})$, after the genealogy and branch times have been integrated out. Calculating the integrals over genealogies and branch times could be achieved by Monte Carlo simulation, but would be computationally intensive. However, as shown by *Meligkotsidou and Fearnhead, 2007*, in one dimension (say $x$) the probability distribution of the locations, $\mathbb{P}(\mathbf{L}|\sigma_x)$, is independent of dispersal, $\sigma_x$, if a scale invariant prior is placed on the branching rate. Intuitively, this follows from the fact that if we do not a priori know the scale of the coalescence rate then we do not know a priori how long the branch lengths are, and so we do not know how fast lineages need to disperse to get to where they are today. Therefore, in that case, conditioning on the locations does not change the likelihood surface of $\sigma_x$ and so can be ignored in the inference of dispersal. This result also holds in two dimensions if we constrain $\sigma_x = \sigma_y$. Sadly, it does not hold in general for an arbitrary dispersal matrix $\boldsymbol{\Sigma}$. For example, if our samples (the end points of a branching Brownian motion) were distributed widely along the $x$ axis but varied little in displacement along the $y$ axis, this would be more likely if $\sigma_y/\sigma_x \ll 1$. Thus, while the overall magnitude of dispersal is not affected by conditioning on the sampling locations, the anisotropy of dispersal may be.

## Composite likelihood over loci

To use information from multiple loci, for eaxmple, to estimate a genome-wide dispersal rate, we can multiply the likelihood ratios together to give a composite likelihood ratio. This is not the full likelihood because it ignores correlations in the genealogical and spatial processes between loci (*Hudson, 2001*; *Larribe and Fearnhead, 2011*; *Varin, 2011*). The maximum likelihood of genome-wide parameters from composite likelihood are known to be statistically consistent in the presence of such correlations (*Wiuf, 2006*), but the likelihood surface is overly peaked due to ignoring the correlations in the data. However, in practice we try to take loci that are far enough apart that their trees are only weakly correlated more than a few tens of generations back.

## Maximum likelihood estimates

The likelihood of the dispersal rate given the shared times and sample locations in a tree is $\mathbb{P}(\mathbf{L}|\mathbf{S}_{\mathcal{G},\mathcal{T}}, \boldsymbol{\Sigma})$. This implies the MLE of the dispersal rate is

$$\widehat{\boldsymbol{\Sigma}} = \frac{(\mathbf{L} - \mathbf{1}\widehat{\ell}_A)^{\mathsf{T}} \mathbf{S}_{\mathcal{G},\mathcal{T}}^{-1} (\mathbf{L} - \mathbf{1}\widehat{\ell}_A)}{n}. \tag{13}$$

This estimate is, however, biased because it relies on the unknown location of the root of the tree. Using the mean-centred likelihood, $\mathbb{P}(\mathbf{X}|\mathbf{V}_{\mathcal{G},\mathcal{T}}, \boldsymbol{\Sigma})$, the restricted MLE of the dispersal rate given for subtree $j$ is

$$\widehat{\boldsymbol{\Sigma}}^* = \frac{\mathbf{X}^{\mathsf{T}} \mathbf{V}_{\mathcal{G},\mathcal{T}}^{-1} \mathbf{X}}{n-1}, \tag{14}$$

which is unbiased and equivalent to $n\widehat{\boldsymbol{\Sigma}}/(n-1)$.

When we chop a tree into subtrees and multiply the (mean-centred) likelihoods over subtrees, the (restricted) maximum likelihood dispersal rate becomes a weighted average of the (restricted) maximum likelihood dispersal rates from each subtree,

$$\widehat{\boldsymbol{\Sigma}}^* = \frac{\sum_j (n_j - 1)\widehat{\boldsymbol{\Sigma}}_j^*}{\sum_j (n_j - 1)}, \tag{15}$$

where $\widehat{\boldsymbol{\Sigma}}_j^*$ is the (restricted) maximum likelihood dispersal rate from subtree $j$ (calculated as above) and $n_j$ is the number of samples in subtree $j$.

Similarly, when we then also multiply (mean-centred) likelihoods across loci the (restricted) maximum likelihood dispersal rate is a weighted average of the (restricted) estimates over the subtrees at all loci,

$$\widehat{\boldsymbol{\Sigma}}^* = \frac{\sum_i \sum_j (n_{ij} - 1)\widehat{\boldsymbol{\Sigma}}_{ij}^*}{\sum_i \sum_j (n_{ij} - 1)}, \tag{16}$$

where $\widehat{\boldsymbol{\Sigma}}_{ij}^*$ is the restricted maximum likelihood dispersal rate from subtree $j$ at locus $i$ and $n_{ij}$ is the number of samples in subtree $j$ at locus $i$.

Unfortunately, there is no analytical expression for the maximum likelihood dispersal rate with importance sampling, and so we instead numerically search for the dispersal rate (and branching rate, $\lambda$) which minimizes the negative log composite likelihood ratio (the negative log of the product of *Equation 4* over loci). We perform this search in Python with scipy's optimize.minimize function (*Virtanen et al., 2020*) using the L-BFGS-B method, which allows us to implement bounds (the standard deviations must be non-negative and the correlation must be between −1 and 1). We initialize the search with the BLUP of the dispersal rate (below). To initialize the branching rate we first solve $n(t) = n(0)\exp(\lambda_0 t)$ for $\lambda_0$, giving $\lambda_0 = \ln(n(t)/n(0))/t$, where $n(t)$ is the number of samples, $n(0)$ is the initial number of lineages in a tree, and $t$ is the length of the tree. We then average $\lambda_0$ over all trees and loci.

## Best linear unbiased predictor

There is an analytical approximation to the maximum likelihood dispersal rate when we sample multiple trees per locus but ignore the importance weights (we need to ignore the importance weights because we do not know the branching rate, $\lambda$, without doing a numerical search). Let $\widehat{\boldsymbol{\Sigma}}_{ij}^*$ be the (restricted) maximum likelihood dispersal rate from tree $j$ at locus $i$ (as calculated from *Equation 15*). Averaging this over all trees and loci then gives a BLUP of the (restricted) maximum likelihood dispersal rate. In the main text we numerically search for the (restricted) maximum likelihood dispersal rate, which allows us to include importance weights and more accurately infers simulated dispersal rates, but we implement the BLUP method in our software as it can be a useful way to quickly get a rough estimate of dispersal while still accounting for tree uncertainty. For both simulations and *Arabidopsis*, the BLUP dispersal rate tended to be roughly twice that of the MLE dispersal rate.

## **Ancestor locations**

### The probability distribution of an ancestor locations given a tree and the dispersal rate

A point on a tree represents a genetic ancestor. Choosing one such point, if the ancestor's lineage shares $\mathbf{s}_a$ time with each of the sample lineages and $s_a$ time with itself, then, by the properties of the conditional normal distribution, the probability that the ancestor was at location $\boldsymbol{\ell}_a$ is normally distributed around

$$\widehat{\boldsymbol{\ell}}_a = \widehat{\boldsymbol{\ell}}_A + \mathbf{s}_a^{\mathsf{T}} \mathbf{S}_{\mathcal{G},\mathcal{T}}^{-1} (\mathbf{L} - \mathbf{1}\widehat{\boldsymbol{\ell}}_A) \tag{17}$$

with covariance

$$\widehat{\mathbf{S}} = (s_a - \mathbf{s}_a^{\mathsf{T}} \mathbf{S}_{\mathcal{G},\mathcal{T}}^{-1} \mathbf{s}_a)\boldsymbol{\Sigma}. \tag{18}$$

## Mean centring

This approach gives the correct mean (*Equation 17*) but the uncertainty (*Equation 18*) is incorrect because the approach implicitly assumes we know with certainty where the most recent common ancestor was.

The correct uncertainty is derived by mean centring. The mean-centred sample locations, $\mathbf{X}$, and shared times among the samples, $\mathbf{V}_{\mathcal{G},\mathcal{T}}$, are described above. The mean-centred vector of shared times between the ancestor and the samples is found by calculating the covariances between their mean-centred locations, giving $\mathbf{v}_a = \mathbf{M}(\mathbf{s}_a - \bar{\mathbf{s}}_i)$, where $M$ is the mean-centring matrix (defined above) and $\bar{\mathbf{s}}_i$ is the mean shared time of each sample over all others. The mean-centred shared time of the ancestor with itself is found by calculating the variance in its mean-centred location, giving $v_a = s_a - 2\bar{\mathbf{s}}_a + \bar{\mathbf{s}}$, where $\bar{\mathbf{s}}_a$ is the mean shared time of the ancestor over all samples and $\bar{\mathbf{s}}$ is the average over all elements of $\mathbf{S}_{\mathcal{G},\mathcal{T}}$.

The probability the ancestor was at location $\ell_a$ is then normally distributed with mean as above but now written

$$\widehat{\ell_a} = \overline{\mathbf{L}} + \mathbf{v}_a^\mathsf{T} \mathbf{V}_{\mathcal{G},\mathcal{T}}^{-1} \mathbf{X} \tag{19}$$

and covariance

$$\widehat{\mathbf{S}}^* = \left( v_a - \mathbf{v}_a^\mathsf{T} \mathbf{V}_{\mathcal{G},\mathcal{T}}^{-1} \mathbf{v}_a \right) \mathbf{\Sigma}, \tag{20}$$

where $\overline{\mathbf{L}}$ is the mean location of the samples.

This covariance gives the correct uncertainty. For example, it produces a linear increase in the variance of the location as we move up from a sample when there is only a single sample, that is, this is simple Brownian motion. When there are two samples with a recent common ancestor at time $t$, the variance in ancestor location along this tree back to the most recent common ancestor is maximized at the most recent common ancestor, $t/2$, that is, this is a Brownian bridge. And so on.

## Chopping the tree

As with the dispersal likelihood, to reduce dependencies on distant times we can chop the tree at some time $T$ and use only the subtree containing the ancestor of interest. Often the most recent common ancestor of the resulting subtree occurs more recently than $T$ but we want to infer the location all the way back to $T$, and so we need to infer the location of the common ancestral lineage past the most recent common ancestor. Fortunately, this method extends naturally to times beyond the most recent common ancestor, implicitly modelling simple Brownian motion (a fixed mean and linearly increasing variance) up the stem of the tree. Similarly, the ancestor need not have descendants in the sample – our method implicitly adds simple Brownian motion down the branch leading to a 'hanging' ancestor (which is useful when we want to ignore the location of a sample and locate it or its ancestors using the remaining sample locations and the trees). When there is only a single sample in a subtree we cannot mean centre and instead model simple Brownian motion up from the sample.

## Importance sampling

Finally, to integrate over tree uncertainty and reduce bias from the way in which we sample trees we importance sample. We do this by replacing the dispersal likelihood, $\mathbb{P}(\mathbf{L}|\mathcal{G}, \mathcal{T}, \mathbf{\Sigma})$, in *Equation 4* with the probability distribution for the ancestor's location, $\mathbb{P}(\ell_a|\mathbf{X}, \mathbf{V}_{\mathcal{G},\mathcal{T}}, \mathbf{\Sigma}, \mathbf{v}_a, v_a)$,

$$\widehat{\mathrm{LR}(\ell_a)} = \frac{1}{M} \sum_{m=1}^{M} \frac{\mathbb{P}(\ell_a|\mathbf{X}, \mathbf{V}_{\mathcal{G},\mathcal{T}_m}, \mathbf{\Sigma}, \mathbf{v}_a, v_a)\mathbb{P}(\mathcal{T}_m|\mathcal{G}, \mathbf{\Sigma})}{\mathbb{P}(\mathcal{T}_m|\mathcal{G}, \mathrm{panmixia})}. \tag{21}$$

We compute this using the maximum composite likelihood ratio estimates of $\mathbf{\Sigma}$ and $\lambda$.

## Best linear unbiased predictor

As with dispersal, the above approach requires a numerical search for the most likely ancestor location because a sum of normal distributions (*Equation 21*) does not, in general, follow a tractable distribution. An alternative approach, however, is to calculate the most likely ancestor location for each

sampled tree (*Equation 19*) and importance sample over these estimates. Writing the most likely ancestor location given a sampled tree (*Equation 19*) as $\widehat{\ell}_a(\mathcal{T}_m, \mathcal{G})$, we then have another estimate of the ancestor's location,

$$\tilde{\ell}_a = \frac{1}{M} \sum_{m=1}^{M} \frac{\widehat{\ell}_a(\mathcal{T}_m, \mathcal{G}) \mathbb{P}(\mathcal{T}_m | \mathcal{G}, \mathbf{\Sigma})}{\mathbb{P}(\mathcal{T}_m | \mathcal{G}, \text{panmixia})}. \tag{22}$$

*Equation 22* is a BLUP averaged over the importance weights. We can calculate this directly and it is therefore, in principle, faster to compute than the search over *Equation 21*. It is also independent of the dispersal rate since $\mathbb{P}(\mathcal{T}_m | \mathcal{G}, \mathbf{\Sigma}) = \mathbb{P}(\mathcal{T}_m | \mathcal{G})$ under our spatial model. In practice we find that the two estimators are very correlated, and the more correct numerical search over *Equation 21* is reasonably quick. Throughout the paper we use the numerical search over *Equation 21* but the BLUP method is also implemented in our software. Our software also allows one to importance sample over ancestor location uncertainty, replacing $\widehat{\ell}_a(\mathcal{T}_m, \mathcal{G})$ with the variance in $\ell_a(\mathcal{T}_m, \mathcal{G})$ in *Equation 22*.

## Simulations

Our full pipeline is described in detail in a Snakemake (*Mölder et al., 2021*) file at https://github.com/mmosmond/spacetrees-ms, copy archived at *Osmond, 2024a*.

### Spatially explicit forward-time simulations

We performed simulations in SLiM v4.0.1 (*Haller and Messer, 2023*) with tree-sequence recording (*Haller et al., 2019*). The SLiM code was adapted from the pyslim spatial vignette (https://tskit.dev/pyslim/docs/stable/vignette_space) to model non-overlapping generations.

Individuals are diploid for a single chromosome with $L = 10^8$ basepairs and per basepair recombination rate $r = 10^{-8}$. Individuals exist in a two-dimensional habitat, a square of width $W = 100$ with reflecting boundaries.

Each generation begins with reproduction. Each individual acts once as a 'mother' and chooses a 'father' at random (individuals are hermaphrodites), weighted by their mating weights. The mating weight of an individual distance $d$ from a mother follows a two-dimensional normal distribution centred on the mother with variance $\sigma_m^2 = 1/4$ in both directions and no covariance. Individuals further apart than $3\sigma_m$ are ignored for efficiency.

Local density dependence is modelled through competitive effects on fecundity. The strength of competitive interaction between two individuals distance $d$ distance apart follows a two-dimensional normal distribution centred on one of the individuals with variance $\sigma_c^2 = 1$ in both directions and no covariance. We again ignore individuals more distant than $3\sigma_c$. The number of offspring produced by a mother is Poisson, with mean $R/(1 + C/K)$, where $C$ is the sum of interaction strengths the mother experiences, $R = 2$ is the mean number of offspring in the absence of competition, and $K = 1$ is the local carrying capacity. Each offspring disperses from its mother by a random two-dimensional normal deviate with variance $\sigma_d^2$ in each dimension and no covariance.

Note that, given a lineage has an equal chance of descending from its mother or its father each generation, the true simulated dispersal rate is $\sigma^2 = \sigma_d^2/2 + (\sigma_d^2 + \sigma_m^2)/2 = \sigma_d^2 + \sigma_m^2/2$ (*Smith et al., 2023*) in each dimension.

After reproduction all adults die and all offspring become adults in the next generation. We begin the population with $N_0 = W^2 K$ individuals distributed uniformly at random across space. We end the simulation, and output the tree sequence, after $4N_0$ generations.

### True tree sequence of the sample

Using tskit v0.5.6 (*Kelleher et al., 2018*), we load the tree sequence into Python v3.11.5, sample $k/2 = 50$ present day diploid individuals ($k = 100$ genomes) at random, and simplify the tree sequence to lineages ancestral to the sample. We then use pyslim v0.6 (https://github.com/tskit-dev/pyslim) (copy archived at *Ralph et al., 2024*) to 'recapitate' the tree sequence under the coalescent with recombination (*Hudson, 2002*) with effective population size $N_0$, ensuring all sampled lineages have coalesced. This tree sequence represents the true genealogical history of the sample.

## Inferred tree sequence of the sample

Because we will not know the true tree sequence of any natural sample, we next use msprime v1.3.1 (**Kelleher et al., 2016a**) to layer neutral mutations on the tree sequence with per basepair per generation mutation rate $U = 10^{-8}$. We then write the genotypic data out as a VCF and use Relate v1.2.1 (**Speidel et al., 2019**) to infer the tree sequence. We give Relate the true uniform recombination map and mutation rate and an initial effective population size of $\theta/(4U)$, where $\theta$ is the observed mean genetic diversity calculated from the tree sequence with tskit (**Ralph et al., 2020**). We then feed the resulting output to Relate's 'EstimatePopulationSize' function to iteratively estimate a piecewise-constant effective population size and branch lengths, using five MCMC iterations (the default) and dropping 50% of the trees (the default).

## Processing the tree sequence

At every 100th tree we use Relate's 'SampleBranchLengths' function to sample branch lengths from the posterior $M = 1000$ times, using the true mutation rate and saving the output in Newick format. We next load the Newick trees into Python as one-tree tskit tree sequences with tsconvert (https://github.com/tskit-dev/tsconvert) (copy archived at **Jefferyn et al., 2024**) and calculate the shared times and coalescent times. We then chop these times at a cutoff $T$ and calculate various quantities that are used for inference: the shared time matrix of each subtree, the inverted mean-centred shared time matrix of each subtree, the log of the determinant of the mean-centred shared time matrix of each subtree, the coalescence times, and the log probability of the coalescence times. This preprocessing speeds up the numerical search for maximum likelihood parameters.

## Dispersal estimates

We approximate the maximum composite likelihood estimate of the dispersal rate by numerically searching for the dispersal rate (and branching rate, $\lambda$) that minmizes the negative log of the product of likelihood ratios (**Equation 4**) over sampled loci. We use the L-BFGS-B method of SciPy v1.11.2 (**Virtanen et al., 2020**) to numerically find the maximum, which allows us to impose bounds on the parameters. We search for the maximum in terms of the standard deviation of dispersal in $x$ and $y$ (with a lower bound of $10^{-6}$ to prevent negative values) and the correlation between these two axes (with bounds of $-0.99$ and $0.99$ to prevent estimates with absolute value greater than 1). We also place a lower bound of $10^{-6}$ on $\lambda$ to prevent negative branching rates.

## Locating genetic ancestors

To locate a genetic ancestor at a particular locus and time we numerically search for the location that maximizes the likelihood ratio (**Equation 21**), following the same approach as above but with the SLSQP method without bounds.

## A. thaliana data

Our full pipeline is described in detail in a Snakemake (**Mölder et al., 2021**) file at https://github.com/mmosmond/spacetrees-ms (copy archived at **Osmond, 2024a**).

## Inferring the tree sequence

We first downloaded 63 fastq sequences from https://www.ebi.ac.uk/ena/browser/view/PRJEB55632 and https://www.ebi.ac.uk/ena/browser/view/PRJEB55353 (**Wlodzimierz et al., 2023**) and 3 genome assemblies: Col-0 and Ey15-2 from https://www.ncbi.nlm.nih.gov/bioproject/PRJEB50694 and Kew-1 from https://www.ncbi.nlm.nih.gov/bioproject/PRJEB51511. We also downloaded a hard-masked TAIR10 reference genome from https://zenodo.org/records/7326462.

To extract the sample haplotypes from this data we roughly followed the methods of **Wlodzimierz et al., 2023**. We first used hifiasm v0.19.5 (**Cheng et al., 2021**; **Cheng et al., 2022**) to assemble each of the 63 fastqs into contigs and convert the outputs to fasta. We then used ragtag v2.1.0 (**Alonge et al., 2022**), with the minimap2 v2.26 (**Li, 2018**; **Li, 2021**) aligner and the TAIR10 reference, to scaffold each of the 63 assemblies. We next extracted the 5 autosomes from each of the 66 assemblies, aligned these to the TAIR10 reference with minimap2 and converted to sam/bam format with samtools v1.13 (**Danecek et al., 2021**). We then used bcftools v1.13 (**Danecek et al., 2021**) and the

TAIR10 reference to create a VCF for each chromosome. We also used bcftools to filter out sites with missing or heterozygote genotypes and filter to just biallelic single-nucleotide polymorphisms (SNPs). To get the haplotypes of each sample we used bcftools to convert the VCFs to hap/sample format and from this kept just one haplotype of each (now explicitly haploid) sample.

To polarize the SNPs we used a multispecies alignment for *A. thaliana*, *Boechera stricta*, *Arabidopsis lyrata*, and *Malcomia maritima* (available at https://doi.org/10.5281/zenodo.11456353). We used bx-python v0.8.9 (https://github.com/bxlab/bx-python; *Taylor et al., 2020*) to create a fasta file containing the three outgroups from this alignment and, combined with the hap files described above, created the input files required by est-sfs (*Keightley and Jackson, 2018*). We then ran est-sfs v2.04, separately for each chromosome, using the Kimura 2-parameter model (*Kimura, 1980*) to get the probability that each major allele is ancestral. This method includes information about polymorphism within the ingroup when assigning ancestral states. Wherever the reference allele has a less than 1/2 probability of being ancestral we 'flip' the alleles at that locus, so that 0 and 1 refer to ancestral and derived alleles, respectively, in our hap files.

To remove difficult to sequence regions of the genome we downloaded the hard-masked TAIR10 reference from https://ftp.ensemblgenomes.ebi.ac.uk/pub/plants/release-56/fasta/arabidopsis_thaliana/dna/ and use Relate to mask the hap files. We also downloaded a recombination map from https://popsim-consortium.github.io/stdpopsim-docs/stable/catalog.html#sec_catalog_aratha_genetic_maps_salomeaveraged_tair10 (*Adrion et al., 2020*; *Salomé et al., 2012*) and converted it into the form required by Relate.

We ran Relate v1.1.9 with the polarized, masked, haploid hap files and the recombination map to infer the genome-wide genealogies, with initial effective population size $10^5$ and mutation rate $7 \times 10^{-9}$(*Adrion et al., 2020*). We then used Relate to infer a piecewise-constant effective population size and re-estimate branch lengths accordingly, using 10 MCMC iterations (the default is 5) and 50% of the trees (default). The final anc/mut output is converted into a tskit tree sequence with Relate for additional analyses. We also use Relate to sample branch lengths 1000 times at every 100th tree on each chromosome (for a total of 878 trees).

To extract the information we need from the Newick files output by Relate we used tsconvert v0.1 (https://github.com/tskit-dev/tsconvert) (copy archived at *Jefferyn et al., 2024*) to convert them into (one-tree) tskit tree sequences and tskit v0.5.2 to extract the shared times and coalescent times from each. We then applied the time cutoffs, and for each subtree pre-computed the inverse and determinant of the mean-centred shared time matrix. We also pre-calculated the probability of the coalescence times given the piecewise-constant effective population size inferred by Relate. With these quantities in hand we estimate dispersal rates and locate genetic ancestors.

## Dispersal estimates

We estimated dispersal rates as described above for simulations. We converted our estimates of dispersal rate from degrees squared to kilometres squared by multiplying the latitudinal estimates by $111^2$ and by multiplying the longitudinal estimates by $(111 \cos(\bar{y}\pi/180))^2$, where $\bar{y}$ is the mean latitude across samples.

As one of our results is that rates of dispersal are higher along latitude than along longitude, we wanted to make sure that the lack of conditioning on sampling location did not drive this result. To do this, we ran our method looking at only longitudinal dispersal, ignoring the sample latitudes, and then again looking at only latitudinal dispersal, ignoring the sample longitudes. This reduces the problem back to one dimension where the conditioning does not matter (as discussed above and in *Meligkotsidou and Fearnhead, 2007*). Doing this we find essentially identical dispersal rates as before, justifying our result of a larger latitudinal dispersal rate despite not conditioning on the sample locations.

## Locating genetic ancestors

We locate ancestors as described above for simulations, but without imposing any bounds.

## Acknowledgements

We thank Tyler Kent, Adrian Platts, and the Brassicales Map Alignment Project (DOE-JGI, http://bmap.jgi.doe.gov/) for the multispecies genome alignments, Yan Wong, Ben Haller, and Peter Ralph

for the retainCoalescentOnly and permanent updates to SLiM/pyslim, Leo Speidel for help with Relate, Vince Buffalo for an introduction to Snakemake, Fabrizio Menardo for pointing out an error in our lat/long to km conversion, Puneeth Deraje and James Kitchens for helpful discussions, and Nick Barton for healthy skepticism about per-locus estimates of ancestor locations at deep times. Funding provided by Banting (Canada), the Center for Population Biology (UC Davis), the Natural Sciences and Engineering Research Council of Canada (RGPIN-2021-03207, awarded to MO), and the National Institute of General Medical Sciences of the National Institutes of Health (NIH R01 GM108779 and R35 GM136290, awarded to GC). Computations were performed on the Niagara supercomputer at the SciNet HPC Consortium. SciNet is funded by: the Canada Foundation for Innovation; the Government of Ontario; Ontario Research Fund – Research Excellence; and the University of Toronto.

## Additional information

### Competing interests

Graham Coop: Reviewing editor, eLife. The other author declares that no competing interests exist.

### Funding

| Funder | Grant reference number | Author |
| --- | --- | --- |
| Natural Sciences and Engineering Research Council of Canada | RGPIN-2021-03207 | Matthew Osmond |
| Natural Sciences and Engineering Research Council of Canada | DGECR-2021-00114 | Matthew Osmond |
| Banting Research Foundation | | Matthew Osmond |
| National Institute of General Medical Sciences | R01 GM108779 | Graham Coop |
| National Institute of General Medical Sciences | R35 GM136290 | Graham Coop |

The funders had no role in study design, data collection, and interpretation, or the decision to submit the work for publication.

### Author contributions

Matthew Osmond, Conceptualization, Resources, Data curation, Software, Formal analysis, Funding acquisition, Validation, Investigation, Visualization, Methodology, Writing – original draft, Writing – review and editing; Graham Coop, Conceptualization, Formal analysis, Supervision, Funding acquisition, Methodology, Writing – original draft, Writing – review and editing

### Author ORCIDs

Matthew Osmond  https://orcid.org/0000-0001-6170-8182
Graham Coop  https://orcid.org/0000-0001-8431-0302

### Decision letter and Author response

Decision letter https://doi.org/10.7554/eLife.72177.sa1
Author response https://doi.org/10.7554/eLife.72177.sa2

## Additional files

### Supplementary files

• MDAR checklist

## Data availability

All code used to perform the analyses in this study can be found at https://github.com/mmosmond/spacetrees-ms (copy archived at *Osmond, 2024a*). More information on how to run our method, spacetrees, is available at https://github.com/osmond-lab/spacetrees (copy archived at *Osmond, 2024b*).

The following dataset was generated:

| Author(s) | Year | Dataset title | Dataset URL | Database and Identifier |
|---|---|---|---|---|
| Osmond MM, Coop GC | 2024 | Relate-inferred genealogies for 66 longread *Arabidopsis thaliana* genomes | https://doi.org/10.5281/zenodo.5099656 | Zenodo, 10.5281/zenodo.5099656 |

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
