## [Editor Report]

This fundamental and pioneering paper demonstrates the power of using the Ancestral Recombination Graph in estimating historical dispersal rates and illustrates the importance of using good data. The methodology is compelling and well beyond the state-of-the-art. The paper should be of interest to anyone working with population genetic inference.

---

## [Decision Letter]

**Decision letter after peer review:**

Thank you for submitting your article "Estimating dispersal rates and locating genetic ancestors with genome-wide genealogies" for consideration by *eLife*. Your article has been reviewed by 3 peer reviewers including Magnus Nordborg as the Reviewing Editor and Reviewer #1, and the evaluation has been overseen by Detlef Weigel as the Senior Editor. The following individuals involved in review of your submission have agreed to reveal their identity: Leo Speidel (Reviewer #2); Jerome Kelleher (Reviewer #3).

The reviewers have discussed their reviews with one another at length, and the Reviewing Editor has drafted this to help you prepare a revised submission.

Essential revisions (for the authors):

Apologies for the delayed decision, which is almost entirely due to the Reviewing Editor being overcommitted. All reviewers agreed that this is a technically excellent paper that should be of general interest to anyone working with polymorphism data. Contrary to standard *eLife* practice, the individual reviews are included below, as they are all clear and contain numerous small fixes you will want to implement.

We have only one serious concern, which is that the time estimates from the Arabidopsis data may be seriously biased, and by serious, we mean an order of magnitude or two. Rapid recent migration (hundreds of kilometer in tens of generations) is exceedingly hard to reconcile with what is known about Arabidopsis biology, and probably also with standard population genetics models of gene flow and drift.

After extensive discussions (Magnus Nordborg also wishes to acknowledge lab members, in particular Anna Igolkina), we suspect the 5% allele frequency cut-off may be to blame. If the age of an allele is roughy proportional to its frequency, a 5% cut-off in Arabidopsis could pretty much erase all post-glacial history. Which would work: your estimates would be plausible if you were talking of thousands rather than tens of generations.

Other possible biases include the use of imputed and heavily filtered data (see comments by Reviewer #1), but this is probably less important.

We would strongly encourage you to examine so of the inferred trees manually, doing some back-of-the-envelope calculations using the full data. Nordborg and Weigel would also happily share data from unpublished de novo PacBio assemblies, which would let you look at the full polymorphism spectrum on local trees.

As this is a method that is likely to set a new standard for population genetic data analysis, we feel that it is very important to get the application to real data right.

*Reviewer #1 (Recommendations for the authors):*

Unless I'm very much mistaken, something is going wrong, and I hope we can figure out what it is.

Others are better suited to comment on the algorithm than I, but here are a few concerns about the data:

First, you used the imputed SNP data. I wouldn't. We should have made this clearer in the paper, but imputation was only done because various silly GWAS software couldn't handle missing data. Imputation was never the right thing to do, because the data missing data are not missing because we didn't look, but because we couldn't interpret what we saw. There was adequate coverage, and no reactions failed. SNP calling failed because alignment failed. In other words, most "missing data" probably represents hyper-variable regions and structural variants (TEs, in particular). When these sites are imputed, they are forced to look like more conserved haplotypes, and I think I don't need to explain what this likely does to your analysis…

When doing GWAS, this is not a problem. You simply get a large stretch of identical p-values, which we interpret as "Uh-oh, better go look at the raw data to figure out wtf is going on in this region!". Alternatively, we look at the PacBio genomes (we have over 100 already).

Second, I'm under the impression that the tree-reconstruction algorithm (Speidel et al., 2019) relies on polarized SNPs, and that you get those data from Tyler Kent. Fine, but be aware that this results in a huge bias, because a large fraction of SNPs cannot be polarized. A. lyrata is > 12% divergent, most of the non-coding genome cannot be aligned. Again you are biasing yourself toward conserved regions.

I seriously think you need to do some basic modeling to check whether your results are consistent with other observations. I don't think they are. I'm referring to the time estimates here. I would bet you are off by at least an order of magnitude.

To be discussed, I hope.

*Reviewer #2 (Recommendations for the authors):*

The method:

- The method is described well and the mathematical foundations and technical details are sound. I have no concerns about the way Relate trees are used, e.g., in the importance sampler.

- While overall accuracy of the method is very good, an interesting phenomenon in simulated data is the overestimation of dispersal rates using Relate trees (as compared to true trees). To disentangle causes of this bias, would it be possible to – (A) Infer branch lengths given true tree topologies (by converting *trees files into Relate format, sampling branch lengths, then converting back to *trees format)? If bias comes from errors in tree topologies, there should be no bias when using true tree topologies (and this could also be used to illustrate the usefulness of the importance sampler to correct for the potential bias introduced by the panmixia assumption in the branch length estimator).

- (B) Measure accuracy of inferred tree topologies and filter out inaccurate trees (or simulate data with lower recombination rates where tree topologies should be more accurate)

- In practise, I think it could be worth filtering out trees in recombination hotspots, if this hasn't been done already. E.g., one could think of a strategy of picking a tree every x bases and then filtering out unreliable trees according to some criteria, such as recombination rate. It would be very interesting to see if this makes a difference!

- I couldn't find a mentioning of the sample size used for the simulations in Figure 2. How does the method scale with data size and how does the above bias change?

- Do you think it is possible to partially integrate over topologies by e.g., inferring trees for subsamples and multiplying the likelihoods across these, with the potential downside of loosing some power? While trees won't be independent, the MLE's are I think still unbiased in a composite likelihood.

- I particularly like the idea of "chopping" the trees at a certain height to avoid having to rely on inferred locations at deep times, which I agree are most likely highly unreliable. However, I didn't understand whether the mean centering of locations applied to each subtree separately (I think it does). I was wondering if it is technically possible to improve on this slightly if one was able to predict the average ancestor location with given descendant set and time of the TMRCA (which is less than cutoff T) in the past using genome-wide data.

- In Figure 1, it looks like the ancestor "A", for which we want to infer the geographic location is placed on a local genealogy. However, when sampling branch lengths, the age of this ancestor will vary, so I wasn't sure if this diagram is accurate. Do you instead look at the ancestor of e.g., sample 4 at time x? So this could fall below or above the coalescence with sample 5?

- In Figure 2 E and G, I didn't understand what the solid points represent.

- On page 9, the authors mention that they implemented a BLUP for inferring ancestral locations which performs similarly to their first appraoch. Is there a figure or correlation (or similar) that the authors could cite which confirms this claim? The phrasing is slightly ambiguous about which method the authors then continue to use (they are both technically "MLE-based" right?), but I am guessing it is the former.

Application to *A. thaliana*:

- I had no previous background knowledge on the history of these *A. thaliana* but overall had the impression that the method convincingly demonstrates recent migration patterns as claimed by the authors. A few concerns and questions are listed below:

- On p. 11, the authors state that "from our perspective remove the complications of obtaining phased haplotypes". It wasn't clear to me what this meant specifically – does it mean that most of the genome is homozygous, making phasing trivial and/or that then only one haploid sequence is obtained for each individual? I think it would be good to clarify!

- The idea of locating ancestors of American samples is very interesting. You make the claim that these American samples came perhaps predominantly from southern Germany. In a clean split model, where American *A. thaliana* have a separate history in the last 400 years, is my understanding correct that we have no information about more recent geographic location? Why is it then that they are more closely related to French samples today, also indicated in the continued drift towards France <400 years ago? Is this evidence of continued migration into the Americas?

- Figure 7 is very nice!

- I liked the idea of Figure 9 but found the lines a bit difficult to distinguish. In the text you seem to focus primarily on two samples, so would it be possible to just show those two? Or have a separate map for each sample?

- On p.41, the authors state that they filtered out variants of <5% MAF. This seems potentially problematic as it means no mutations are mapped to the bottom of the trees (and presumably all singletons are excluded), meaning the bottom of these trees are artificially shortened. I think this could potentially explain (in part) the signal of recent increase in dispersion. I would strongly recommend not applying this filter, unless there is a fundamental reason as to why it is necessary.

- On p. 41, l927, it is mentioned that the EstimatePopulationSize script needed customising to allow for haploid samples. I think it should work as is – but I would be interested in knowing what the problem was that you encountered!

*Reviewer #3 (Recommendations for the authors):*

I have only one substantive point, which I think can be easily addressed withsome updates to the text. I feel that the manuscript currently glosses over thedifficulties of population genetic models in a 2D continuum, which I worry willlead to others extending the methods in poorly advised directions. I am surethat the BBM model is a good approximation, especially in the limitedtime-slices in which it's being applied, but it's not clear to me how thisrelates to the classical Wright-Malecot models, or the more recentEtheridge-Barton approach. I think a reasonable approach would be tomention the difficulties like the pain in the torus, cite some of theliterature on solving that problem, and provide a simple argument for why theBBM model is a good approximation in this setting.

---

## [Author Response]

Essential revisions (for the authors):Apologies for the delayed decision, which is almost entirely due to the Reviewing Editor being overcommitted. All reviewers agreed that this is a technically excellent paper that should be of general interest to anyone working with polymorphism data. Contrary to standard eLife practice, the individual reviews are included below, as they are all clear and contain numerous small fixes you will want to implement.

Thank-you very much for the kind words and helpful reviews.

We have only one serious concern, which is that the time estimates from the Arabidopsis data may be seriously biased, and by serious, we mean an order of magnitude or two. Rapid recent migration (hundreds of kilometer in tens of generations) is exceedingly hard to reconcile with what is known about Arabidopsis biology, and probably also with standard population genetics models of gene flow and drift.After extensive discussions (Magnus Nordborg also wishes to acknowledge lab members, in particular Anna Igolkina), we suspect the 5% allele frequency cut-off may be to blame. If the age of an allele is roughy proportional to its frequency, a 5% cut-off in Arabidopsis could pretty much erase all post-glacial history. Which would work: your estimates would be plausible if you were talking of thousands rather than tens of generations.Other possible biases include the use of imputed and heavily filtered data (see comments by Reviewer #1), but this is probably less important.We would strongly encourage you to examine so of the inferred trees manually, doing some back-of-the-envelope calculations using the full data. Nordborg and Weigel would also happily share data from unpublished de novo PacBio assemblies, which would let you look at the full polymorphism spectrum on local trees.As this is a method that is likely to set a new standard for population genetic data analysis, we feel that it is very important to get the application to real data right.

Thank you very much for your careful thoughts here, we agree that it is very important to get the application to real data right. This was the major focus of our revisions and we are now much more confident in the inferred trees, and therefore in our estimates of dispersal and the locations of genetic ancestors.

Instead of just dropping the 5% MAF cutoff and running the same pipeline with the imputed haploid genomes, we first decided to improve our analysis by starting with the underlying SNP data, which allowed us to (1) include both genomes of each individual, (2) add ~400 individuals, including new geographic (Africa) and temporal (herbaria) variation, and (3) do the imputing and filtering ourselves. With this bigger dataset we first removed samples with especially low coverage, filtered to biallelic SNPs, imputed and phased, polarized, masked, adjusted the recombination map to account for inbreeding, and inferred the trees. Unfortunately, after all this we estimated similarly large dispersal rates, even from small spatial clusters of individuals.

It may still be possible to more accurately infer the tree sequence, and hence the dispersal rate via our method, from this large short-read dataset, e.g., using a high quality reference panel to do the imputing. However, we found another way around this issue – inferring the trees from long-read sequences, as you suggest. After seeing the Wlodzimierz et al. 2023 paper, we chose to try out our method on these 66 long-read genomes. While this long-read dataset is much smaller, the samples are well-chosen for our purposes, with relatively broad sampling across geography and ancestry. With this long-read dataset we can avoid imputing altogether. The result is, we think, a more reliable tree sequence, which gives a much more reasonable dispersal estimate. Our new dispersal estimates are roughly 2 orders of magnitude smaller (as you guessed), align with simpler pairwise-distance estimates (some of which rely only on pairwise nucleotide diversity, and so are independent of the inferred trees), and are within reasonable back-of-the-envelope bounds given hypothesized movements in *Arabidopsis thaliana*. Fortunately, we can also visualize many of the ancestral movements that we looked at last time with this smaller dataset. We are very happy to now present this more reliable application of our method, and thank you for pushing us in this direction.

Reviewer #1 (Recommendations for the authors):Others are better suited to comment on the algorithm than I, but here are a few concerns about the data:First, you used the imputed SNP data. I wouldn't. We should have made this clearer in the paper, but imputation was only done because various silly GWAS software couldn't handle missing data. Imputation was never the right thing to do, because the data missing data are not missing because we didn't look, but because we couldn't interpret what we saw. There was adequate coverage, and no reactions failed. SNP calling failed because alignment failed. In other words, most "missing data" probably represents hyper-variable regions and structural variants (TEs, in particular). When these sites are imputed, they are forced to look like more conserved haplotypes, and I think I don't need to explain what this likely does to your analysis…When doing GWAS, this is not a problem. You simply get a large stretch of identical p-values, which we interpret as "Uh-oh, better go look at the raw data to figure out wtf is going on in this region!". Alternatively, we look at the PacBio genomes (we have over 100 already).

In our first attempt at re-analysis, with a larger short-read dataset, we started with the un-imputed genotypes. We first tried to avoid imputing with these genotypes, but after removing sites with missing genotypes (Relate does not allow missing data) there were too few sites to infer a tree sequence. We next tried imputing ourselves, and accounted for the fact that some regions of the genome are hard to sequence by masking (based on TAIR10), which removed roughly 50% of sites. Unfortunately the trees inferred from this data still gave unrealistically large dispersal rates. We then turned to the smaller long-read dataset (as you suggest), where we could avoid imputing altogether.

Second, I'm under the impression that the tree-reconstruction algorithm (Speidel et al., 2019) relies on polarized SNPs, and that you get those data from Tyler Kent. Fine, but be aware that this results in a huge bias, because a large fraction of SNPs cannot be polarized. A. lyrata is > 12% divergent, most of the non-coding genome cannot be aligned. Again you are biasing yourself toward conserved regions.

Yes, Relate does rely on polarized SNPs and we use *A. lyrata* as one of our outgroups (along with the less related *Boechera stricta* and *Malcolmia maritima*) to polarize. However, we use est-sfs (Keightley and Jackson 2018) to polarize, which uses both outgroup information as well within-population allele frequencies. This means that we can still polarize SNPs where the outgroups do not align, albeit potentially less accurately, based on allele frequencies within our sample of *thaliana*. We now mention that est-sfs uses ingroup polymorphism info (lines 851-852).

I seriously think you need to do some basic modeling to check whether your results are consistent with other observations. I don't think they are. I'm referring to the time estimates here. I would bet you are off by at least an order of magnitude.

We totally agree. We think our original dispersal estimates were way too high (meaning many branch lengths were way too short). As discussed above (and in the main text), our new dispersal rates are more than 2 orders of magnitude smaller and align with simpler, tree-free estimates and back-of-the-envelope calculations (lines 256-268, 278-288).

To be discussed, I hope.

Indeed!

Reviewer #2 (Recommendations for the authors):The method:- The method is described well and the mathematical foundations and technical details are sound. I have no concerns about the way Relate trees are used, e.g., in the importance sampler.- While overall accuracy of the method is very good, an interesting phenomenon in simulated data is the overestimation of dispersal rates using Relate trees (as compared to true trees). To disentangle causes of this bias, would it be possible to – (A) Infer branch lengths given true tree topologies (by converting *trees files into Relate format, sampling branch lengths, then converting back to *trees format)? If bias comes from errors in tree topologies, there should be no bias when using true tree topologies (and this could also be used to illustrate the usefulness of the importance sampler to correct for the potential bias introduced by the panmixia assumption in the branch length estimator).- (B) Measure accuracy of inferred tree topologies and filter out inaccurate trees (or simulate data with lower recombination rates where tree topologies should be more accurate)

While these are great suggestions to pinpoint the issue, we have decided to leave this work for another paper, as we think a useful future study would be to look at the accuracy of ARGs inferred from spatial simulations more generally. Here, we simply mention that the upward bias is consistent with both error in topologies and documented underestimates of branch lengths (lines 162-173).

- In practise, I think it could be worth filtering out trees in recombination hotspots, if this hasn't been done already. E.g., one could think of a strategy of picking a tree every x bases and then filtering out unreliable trees according to some criteria, such as recombination rate. It would be very interesting to see if this makes a difference!

Trying to choose the most informative independent trees is a good idea, e.g., weighting by tree span. We do not attempt it here but mention the idea in the main text to inform future work (lines 171-173).

- I couldn't find a mentioning of the sample size used for the simulations in Figure 2. How does the method scale with data size and how does the above bias change?

Apologies. We mentioned the sample size in the methods but not in the main text. We’ve now added the sample size to the caption of figure 2 (50 diploid individuals = 100 haploid genomes). We’ve also performed analyses with fewer (25 diploids) and more (100 diploids) samples to see how sample size affects our estimates. These analyses are provided as supplementary figures in figure 2 and mentioned in the main text (lines 167-169, lines 202-204).

- Do you think it is possible to partially integrate over topologies by e.g., inferring trees for subsamples and multiplying the likelihoods across these, with the potential downside of loosing some power? While trees won't be independent, the MLE's are I think still unbiased in a composite likelihood.

This is an interesting idea, applicable to many methods that use Relate. We do not attempt it but mention it to inform future work (lines 582-584).

- I particularly like the idea of "chopping" the trees at a certain height to avoid having to rely on inferred locations at deep times, which I agree are most likely highly unreliable. However, I didn't understand whether the mean centering of locations applied to each subtree separately (I think it does). I was wondering if it is technically possible to improve on this slightly if one was able to predict the average ancestor location with given descendant set and time of the TMRCA (which is less than cutoff T) in the past using genome-wide data.

Yes, the mean-centering is applied to each subtree separately, which we’ve now made clearer in the methods (lines 528-534). This implies the locations of the MRCAs of each subtree are independent unknowns. It is not clear to us exactly what the suggested improvement is.

- In Figure 1, it looks like the ancestor "A", for which we want to infer the geographic location is placed on a local genealogy. However, when sampling branch lengths, the age of this ancestor will vary, so I wasn't sure if this diagram is accurate. Do you instead look at the ancestor of e.g., sample 4 at time x? So this could fall below or above the coalescence with sample 5?

It is the latter: we ask where the ancestor of sample i is at time x, which could fall on different branches depending on the branch lengths. We now make this clearer in the caption of figure 1.

- In Figure 2 E and G, I didn't understand what the solid points represent.

The solid points in Figure 2 E and G are the locations of the samples that we used to infer ancestral locations, which we now mention in the caption.

- On page 9, the authors mention that they implemented a BLUP for inferring ancestral locations which performs similarly to their first appraoch. Is there a figure or correlation (or similar) that the authors could cite which confirms this claim? The phrasing is slightly ambiguous about which method the authors then continue to use (they are both technically "MLE-based" right?), but I am guessing it is the former.

We now include a supplementary figure showing that the BLUP locations have similar accuracy as the MLE locations where we mention this in the main text (lines 182-184).

We now clarify that we continue to use the ‘likelihood’-based method, rather than the ‘BLUP’ method, both of which are defined in the text (lines 180-184).

Application to *A. thaliana*:- I had no previous background knowledge on the history of these *A. thaliana* but overall had the impression that the method convincingly demonstrates recent migration patterns as claimed by the authors. A few concerns and questions are listed below:- On p. 11, the authors state that "from our perspective remove the complications of obtaining phased haplotypes". It wasn't clear to me what this meant specifically – does it mean that most of the genome is homozygous, making phasing trivial and/or that then only one haploid sequence is obtained for each individual? I think it would be good to clarify!

Yes, most of the genome is homozygous, making phasing easier, which we’ve reworded (lines 235-236). And yes, we use only one haploid sequence per individual, which we now explicitly state in the main text (line 227).

- The idea of locating ancestors of American samples is very interesting. You make the claim that these American samples came perhaps predominantly from southern Germany. In a clean split model, where American *A. thaliana* have a separate history in the last 400 years, is my understanding correct that we have no information about more recent geographic location? Why is it then that they are more closely related to French samples today, also indicated in the continued drift towards France <400 years ago? Is this evidence of continued migration into the Americas?

Our argument here was that, while most of the samples in America are most closely related to the samples in France today (relative to the other samples outside of America), we infer the ancestors of these American samples to be in southern Germany 400 years ago. This could be the result of dispersal/gene flow from Germany to France in the last 400 years combined with the replacement of these German ancestors or gene flow from elsewhere into Germany in the last 400 years. The new dataset does not contain any samples in the Americas and so we do not discuss this in the main text.

- Figure 7 is very nice!- I liked the idea of Figure 9 but found the lines a bit difficult to distinguish. In the text you seem to focus primarily on two samples, so would it be possible to just show those two? Or have a separate map for each sample?

Our new `windrose’ figure (now Figure 4) has just one sample per map, for clarity.

- On p.41, the authors state that they filtered out variants of <5% MAF. This seems potentially problematic as it means no mutations are mapped to the bottom of the trees (and presumably all singletons are excluded), meaning the bottom of these trees are artificially shortened. I think this could potentially explain (in part) the signal of recent increase in dispersion. I would strongly recommend not applying this filter, unless there is a fundamental reason as to why it is necessary.

Thank you very much for catching this error. When we analyzed the larger short-read dataset we did not apply this filter but found similarly high dispersal rates, suggesting this was not the source of the problem, but rather bad imputation. In our new analysis we do not filter out sites with rare alleles or impute.

- On p. 41, l927, it is mentioned that the EstimatePopulationSize script needed customising to allow for haploid samples. I think it should work as is – but I would be interested in knowing what the problem was that you encountered!

In our re-analysis we did not edit any Relate files (except to avoid printing with R). We unfortunately do not recall (and can’t find) what lines we commented out in EstimatePopulationSize.sh in our original analysis (near the end if memory serves, Relate v1.1.4).

Reviewer #3 (Recommendations for the authors):I have only one substantive point, which I think can be easily addressed withsome updates to the text. I feel that the manuscript currently glosses over thedifficulties of population genetic models in a 2D continuum, which I worry willlead to others extending the methods in poorly advised directions. I am surethat the BBM model is a good approximation, especially in the limitedtime-slices in which it's being applied, but it's not clear to me how thisrelates to the classical Wright-Malecot models, or the more recentEtheridge-Barton approach. I think a reasonable approach would be tomention the difficulties like the pain in the torus, cite some of theliterature on solving that problem, and provide a simple argument for why theBBM model is a good approximation in this setting.

We have done just this at the end of our methods overview (lines 136-144).